# Bicriteria Algorithms for Submodular Cover with Partition and Fairness Constraints

## Abstract

In many submodular optimization applications, datasets are naturally partitioned into disjoint subsets. These scenarios give rise to submodular optimization problems with partition-based constraints, where the desired solution set should be in some sense balanced, fair, or resource-constrained across these partitions. While existing work on submodular cover largely overlooks this structure, we initiate a comprehensive study of the problem of Submodular Cover with Partition Constraints (SCP) and its key variants. Our main contributions are the development and analysis of scalable bicriteria approximation algorithms for these NP-hard optimization problems for both monotone and nonmonotone objectives. Notably, the algorithms proposed for the monotone case achieve optimal approximation guarantees while significantly reducing query complexity compared to existing methods. Finally, empirical evaluations on real-world and synthetic datasets further validate the efficiency and effectiveness of the proposed algorithms.

## 1 Introduction

Submodular optimization algorithms have emerged as a cornerstone of modern machine learning, driving advancements across a range of impactful applications. From curating high-quality pretraining and fine-tuning datasets for large language models (Ji et al., 2024; Kumari et al., 2024; Agarwal et al., 2024) to powering diversified online recommendation systems (Hiranandani et al., 2020; Chen and Crawford, 2025), multi-agent optimization in robotics (Zhou and Tokekar, 2022; Xu and Tzoumas, 2024), and enabling precise image attribution in computer vision (Chen et al., 2024a). Submodular functions informally satisfy a diminishing returns property that is exhibited by many objective functions for fundamental optimization problems in machine learning. Formally, let $f : 2^U \to \mathbb{R}$ be defined over subsets of a ground set $U$ of size $n$. Then the function $f$ is *submodular* if for all $A \subseteq B \subseteq U$ and $x \notin B$, $f(A \cup \{x\}) - f(A) \geq f(B \cup \{x\}) - f(B)$. Further, $f$ is *monotone* if $f(Y) \geq f(X)$ for every $X \subseteq Y \subseteq U$.

The submodular cover (SC) problem is an important optimization problem with a variety of applications (Iyer and Bilmes, 2013; Chen and Crawford, 2024; Crawford et al., 2019; Mirzasoleiman et al., 2015). In the classical form, the goal of submodular cover is to find a subset $S \subseteq U$ of minimum cost such that $f(S) \geq \tau$, where the cost function is typically cardinality or some additive cost. Existing results on SC take advantage of its relationship with submodular maximization (Iyer and Bilmes, 2013; Chen and Crawford, 2024), which is to find $\arg\max\{f(S) : c(S) \leq \kappa\}$. For example, (Chen and Crawford, 2024; Iyer and Bilmes, 2013) proposed converting algorithms that could convert any bicriteria algorithm for submodular maximization to an algorithm for SC. In particular, an $(\alpha, \beta)$-bicriteria approximation algorithm for the SC problem returns a solution $X$ such that $|X| \leq \alpha|OPT|$ and $f(X) \geq \beta\tau$.

However, a significant limitation of these classical formulations is their inability to model critical applications where the ground set $U$ is partitioned into disjoint groups $U_1, \ldots, U_N$, and the objective is to find a subset that has a budget within each partition, or alternatively is balanced or fair across the partitions. We further illustrate the submodular cover with partition constraints setting with several applications. In video summarization (Mirzasoleiman et al., 2018), the elements of $U$ are frames that are each associated with one of $N$ consecutive regions of time in the video. A submodular function $f$ is formulated to measure how effectively a subset of frames $X$ summarizes the entire video $U$ (Tschiatschek et al., 2014). The goal is to find a subset of frames that is a sufficiently

good summary, i.e. $f(X) \geq \tau$, while limiting the proportion of frames from each time region in the solution, i.e., $c(X \cap U_i) \leq p_i v$ where $p_j \in [0, 1]$ and $v$ is the budget on the cost. As a second example, consider influence maximization (Tschiatschek et al., 2014), where the ground set of users $U$ may be divided into $N$ partitions, reflecting demographics information such as language. In order to choose a subset with balanced distribution across different partitions, we enforce the fairness constraint where $p_j|S| \leq |S \cap U_j| \leq q_j|S|$. Then the objective is to find a fair solution with minimum cardinality such that $f(S) \geq \tau$. Many further applications exist in the literature, including neural network pruning (Chen et al., 2025), high-quality data selection for learning (Killamsetty et al., 2021), and data summarization (El Halabi et al., 2020).

Despite the importance and widespread applications of this problem, prior work remains limited. Chen et al. (2025) studied a special case of the submodular cover with constraints defined on partitions, which is the fairness constraints. However, their proposed discrete method attains only a suboptimal approximation ratio, while the continuous approach incurs prohibitively high query complexity. In contrast, our approach achieves optimal bicriteria approximation ratios with significantly lower query complexity on the problem of SC with fairness constraint.

In this work, we study several distinct submodular cover problems with constraints defined on partitions of the universe $U$, including but not limited to the submodular cover with fairness constraints. Our approach follows the general converting framework by developing converting algorithms that can convert submodular maximization algorithms into submodular cover algorithms. In particular, to construct solutions with objective values closer to the target threshold $\tau$, we propose bicriteria algorithms for submodular maximization with partition constraints. Notably, unlike traditional greedy algorithms that add one feasible element at a time based on marginal gain, our method incrementally selects blocks of elements in each round, where each block respects the cost distribution across different partitions to ensure that elements are selected proportionally to the budget cost. This block-greedy strategy is particularly effective in the submodular cover setting, where achieving values close to the threshold $\tau$ may require selecting sets that exceed the feasibility limits of standard submodular maximization algorithms. In particular, our contributions are summarized as follows.

1. In Section 2.1, we study the Submodular Cover with Partition Constraint (SCP) problem of $\arg\min_{S \subseteq U}\{v : f(S) \geq \tau, |S \cap U_j| \leq p_j v, \forall j \in [N]\}$ in the case where $f$ is nonmonotone. We first propose a general converting algorithm to convert any randomized algorithms for the dual problem of Submodular Maximization with Partition constraint (SMP), into an algorithm for SCP. By proposing a bicriteria algorithm for SMP, we can obtain an algorithm for nonmonotone SCP with a bicriteria approximation ratio of $(O(\frac{(1+\alpha)}{\epsilon}), 1/e - \epsilon)$.

2. Section 2.2 addresses the problem of Monotone Submodular Cover with Knapsack Partition Constraints (SCKP), which is to find $\arg\min_{S \subseteq U}\{v : f(S) \geq \tau, c(S \cap U_j) \leq p_j v, \forall j \in [N]\}$. We first develop an algorithm for the dual optimization problem of Submodular Maximization under Knapsack Partition Constraints (SMKP), which adopts the block-greedy structure. By utilizing a converting procedure, we achieve a $(\frac{(1+\alpha)\ln 1/\epsilon}{\ln 2}, 1 - \epsilon)$ bicriteria-approximation ratio for SCKP.

3. Section 2.3 considers the monotone Submodular Cover problem with Fairness Constraint (SCF), which was recently introduced by Chen et al. (2025). The proposed algorithm achieves the nearly optimal approximation ratio of $(\mathcal{O}(\ln(1/\epsilon)), 1 - \epsilon)$. This matches the approximation ratio for the algorithm of Chen et al. (2025), but their method is continuous and requires $\mathcal{O}(\frac{n^2(1+\alpha)\log^2(\frac{n}{\epsilon})\log n}{\epsilon^4 \alpha})$ queries of $f$ while our method only requires a query complexity of $\mathcal{O}(\frac{n \log(n)\kappa \ln(1/\epsilon)}{\epsilon})$.

Finally, we conduct an experimental evaluation of our algorithms for nonmonotone SCP, monotone SCKP, and monotone SCF. Our results demonstrate that our proposed algorithm for nonmonotone SCP achieves a higher function value compared to the baseline algorithms, and SCKP achieves an improvement in the budget of the cost. Additionally, our algorithm for SCF outperforms the other algorithms proposed in Chen et al. (2025) in terms of the solution set size and fairness difference.

## 1.1 RELATED WORK

In the context of submodular maximization, matroid constraints represent a fundamental and well-studied class of feasibility constraints, with partition constraints serving as a class of particularly prominent special case with widespread applications (El Halabi et al., 2020; Chen et al., 2025). Therefore, algorithms for submodular maximization with a general matroid constraint, which has been extensively studied, can be employed (Nemhauser et al., 1978; Fisher et al., 1978; Calinescu et al., 2011; Badanidiyuru and Vondrák, 2014; Chekuri and Quanrud, 2019; Buchbinder and Feldman, 2024a). The best known approximation ratio for monotone submodular maximization with a matroid constraint is $1 - 1/e$ (Calinescu et al., 2011; Buchbinder and Feldman, 2018). For the more general maximization of a non-monotone submodular function with a matroid constraint, the best-known hardness result is $0.478$ (Gharan and Vondrák, 2011; Qi, 2024). The algorithm with the current best approximation ratio is a continuous one that achieves $0.401$ (Buchbinder and Feldman, 2024b). The combinatorial algorithm with the best approximation ratio is that of Chen et al. (2024b), which achieves a $0.305 - \epsilon$ approximation guarantee in $O\left(k^5 \log(k)n/\epsilon\right)$ queries of $f$. Partition type of constraints are widely found in submodular optimization applications, but despite this there has been little attention towards algorithms specifically designed for them. An exception is that fairness constraints have recently been of interest (El Halabi et al., 2020; 2023; Chen et al., 2025). El Halabi et al. showed that maximization of a monotone submodular function under a fairness constraint can be converted into monotone SM under a matroid constraint.

The Submodular Maximization under Knapsack Partition Constraints (SMKP) problem, which is the dual problem of the SCKP, is defined as $\arg \max\{f(S) : \sum_{s \in X \cap U_j} c(s) \leq p_j v, \forall j \in [N]\}$. This formulation can be regarded as the generalization of submodular maximization subject to a single knapsack constraint (Amanatidis et al., 2020; Cui et al., 2025) in the case where there is only one partition in the universe. Recent work has also studied submodular maximization under both knapsack and partition constraints (Cui et al., 2024; Li et al., 2025). Specifically, these papers address fairness-aware submodular maximization subject to: (i) a knapsack constraint on the total cost of the selected subset across all partitions, and (ii) cardinality constraints within each partition to ensure fair representation. In contrast, SMKP enforces per-partition knapsack constraints (as opposed to a single global knapsack constraint), without imposing cardinality requirements.

In the classical submodular cover problem with integral-valued objective functions, the standard greedy algorithm—which repeatedly selects the element with the highest marginal gain until the objective reaches a threshold $\tau$—achieves an approximation ratio of $O(\log \max_{e \in U} f(e))$ (Wolsey, 1982). For real-valued submodular functions, a common modification is to stop once the function value reaches $(1 - \epsilon)\tau$, yielding algorithms with a $(\ln(1/\epsilon), 1 - \varepsilon)$-bicriteria approximation ratio (Krause et al., 2008; Chen and Crawford, 2024). For the Fair Submodular Cover (FSC) (Chen et al., 2025) problem, the discrete algorithm of Chen et al. achieves a bicriteria approximation ratio of $(\mathcal{O}(1/\epsilon), 1 - \epsilon)$ while our algorithm achieves an improved approximation ratio of $(\mathcal{O}(\ln 1/\epsilon), 1 - \epsilon)$, which matches the approximation ratio of the continuous method proposed in Chen et al. but requires much fewer queries.

## 2 ALGORITHMS AND THEORETICAL ANALYSES

We now present the main results of our paper[1]. We first address the general case of not necessarily monotone, submodular cover with a partition constraint in Section 2.1. Next, we consider monotone submodular cover with a partition constraint, and our results apply even for the more general knapsack cost, in Section 2.2. Finally, we consider the more restricted, but with many interesting applications, setting of fair submodular cover in Section 2.3.

Central to all of our results is the novel algorithmic framework proposed for submodular maximization problems that achieves a bicriteria approximation ratio by running greedy in blocks, where each block is a feasible subset. This block-wise greedy strategy departs from prior approaches that focus on the matroid structure of partition constraints. In contrast, our method exploits the intrinsic relationship between partition constraints and cardinality constraints, leading to improved query complexity and approximation ratio. Throughout the paper, we define the marginal gain of adding an element $u \in U$

---

[1]We summarized our results in a table in the appendix. Please refer to Table 1.

to a set $S \subseteq U$ as $\Delta f(S, u) = f(S \cup u) - f(S)$. Besides, $OPT$ is used to refer to the optimal solution to the instance of submodular optimization that should be clear from the context.

## 2.1 Non-Monotone Submodular Cover with Partition Constraints

In this section, we consider the general nonmonotone Submodular Cover with Partition Constraint (SCP) problem, which is to find a set $S \subseteq U$ that solves

$$\min_{S \subseteq U} \quad v$$
$$\text{s.t.} \quad |S \cap U_j| \leq p_j v, \quad \forall j \in [N],$$
$$f(S) \geq \tau.$$

The $v$ represents a budget to allocate over the partitioned sets, which our goal is to minimize while ensuring $f$ is sufficiently high. The $p_j$ represents the desired portion of the budget to allocate to the $j$-th partition. Without loss of generality, we assume $\sum_{j \in [N]} p_j = 1$. If there is only one single partition in the universe, i.e., $N = 1$, then the optimal value of $v$ is $|S|$, and we recover the classic submodular cover problem. To further illustrate the problem definition, consider the example application of video summarization described in Section 1, where frames are grouped by scene or content type and costs are uniform. Then this would mean the objective is to find a solution with a minimum total budget, while maintaining a balanced allocation across different partitions and ensuring that the summary achieves sufficiently high quality.

In our first result, taking advantage of the relationship between submodular cover and submodular maximization, we introduce a converting algorithm, `convert-rand`, that can convert any randomized bicriteria algorithm for nonmonotone Submodular Maximization with Partition matroid constraint (SMP) into a bicriteria algorithm for nonmonotone SCP. In particular, the SMP with an input budget $v$ is defined as $\max\{f(S) : |S \cap U_i| \leq p_i v, \forall i \in [N]\}$. We formally define the notion of bicriteria approximation for both SMP and SCP in the following.

**Definition 2.1.** An $(\alpha, \beta)$-approximation algorithm for SMP with input budget $v$ returns a solution $X$ that satisfies

$$f(X) \geq \alpha f(OPT),$$
$$|X \cap U_j| \leq \beta p_j v,$$

where $OPT$ is the optimal solution of SMP, i.e., $OPT := \arg\max\{f(S) : |S \cap U_i| \leq p_i v, \forall i \in [N]\}$.

**Definition 2.2.** An $(\alpha, \beta)$-approximation algorithm for SCP returns a solution $X$ with objective value $v_X$ that satisfies

$$v_X \leq \alpha v_{OPT},$$
$$|X \cap U_j| \leq p_j v_X$$
$$f(X) \geq \beta \tau.$$

Here $OPT$ is the optimal solution of SCP, i.e., $OPT := \arg\min\{v : |S \cap U_j| \leq p_j v, \forall j \in [N], f(S) \geq \tau\}$. $v_{OPT}$ is the optimal value of SCP.

Notice that in Chen and Crawford (2024), they proposed an algorithm to convert randomized submodular maximization algorithms into submodular cover algorithms in the case of monotone submodular objectives. In fact, a key challenge in this setting arises from the need to ensure a high-probability guarantee on the function value $f$ by repeatedly invoking the submodular maximization subroutine and applying concentration inequalities. To reduce the number of oracle queries, the algorithm in Chen and Crawford (2024) applies Markov's inequality and operates on a truncated objective function $f_\tau := \min\{\tau, f\}$ throughout the converting algorithm. However, the assumption that $f_\tau$ is submodular only holds when $f$ is monotone. In contrast, our analysis extends to the non-monotone setting by avoiding the truncated objective and instead employing a more delicate analysis when applying the concentration inequality. Specifically, we analyze the deviation of the random variable $\beta f(OPT_g) - f(S_i)$ where $S_i$ is the output solution set of the randomized subroutine algorithm for SMP.

We now present `convert-rand` and its theoretical guarantees. The pseudocode for `convert-rand` is described in Algorithm 4 in Section B in the supplementary material.

---

**Algorithm 1** `nonmono-bi`

---

1: **Input:** $\epsilon$, partition constraint parameters, total budget $v$
2: **Output:** $S \subseteq U$
3: **for** $i = 1$ to $\frac{2}{\epsilon}$ **do**
4:     **for** $j = 1$ to $N$ **do**
5:         **for** $l = 1$ to $p_j v$ **do**
6:             Let $M \subseteq U_j/S$ be a set of size $\frac{2p_j v}{\epsilon}$ maximizing $\sum_{x \in M} \Delta f(S, x)$.
7:             $u \leftarrow$ uniformly sample an element from $M$
8:             $S \leftarrow S \cup \{u\}$

---

`convert-rand` runs by iteratively guessing the value of the optimal budget $v$. For each guess, `convert-rand` runs the corresponding dual submodular maximization algorithms over multiple independent trials. The theoretical guarantee of `convert-rand` is provided in Theorem 2.3. We defer the analysis to Section B in the supplementary material.

**Theorem 2.3.** *Any randomized $(\gamma, \beta)$-bicriteria approximation algorithm for nonmonotone SMP that runs in time $\mathcal{T}(n)$ where $\gamma$ holds only in expectation can be converted into an approximation algorithm for nonmonotone SCP that with probability at least $1 - \delta$ is a $((1+\alpha)\beta, \gamma - \epsilon)$-bicriteria approximation algorithm that runs in time $O(\log_{1+\alpha}(|OPT|)\ln(1/\delta)\mathcal{T}(n)/\ln(\frac{\beta-\gamma+\epsilon}{\beta-\gamma}))$.*

Since the best-known result of algorithms for nonmonotone submodular maximization over the partition matroid is the one proposed in Chen et al. (2024b), which achieves an approximation ratio of $0.305 - \epsilon$. Therefore by applying Theorem 2.3 to the randomized algorithm in Chen et al. (2024b), we have a $(1 + \alpha, 0.305 - \epsilon)$-bicriteria approximation algorithm for SCP with high probability in $O(n|OPT|\log_{1+\alpha}(|OPT|)\ln(1/\delta)/\ln(1+\epsilon))$ queries of $f$. However, a factor of $0.305 - \epsilon$ of $\tau$ is not very close to a feasible solution, and a natural question arises whether an algorithm that achieves a better feasibility guarantee exists. An important relevant result (Crawford, 2023) is that it has been shown that a feasibility factor better than $1/2$ is impossible for the nonmonotone submodular cover problem. Since this problem is a special case of SCP, this result holds for SCP as well. Still, this leaves us with uncertainty of whether there exist scalable algorithms with approximation ratios in the gap between $0.305$ to $0.5$.

In the rest of this section, we present a scalable algorithm, `nonmono-bi`, that can output a solution set arbitrarily close to $\tau/e$. The pseudocode of `nonmono-bi` is provided in Algorithm 1. `nonmono-bi` uses the idea of gradually developing a solution in blocks greedy algorithm, and achieves the bicriteria-approximation ratio as below.

**Theorem 2.4.** *Suppose that `nonmono-bi` is run for an instance of nonmonotone SMP with budget $v$, then `nonmono-bi` outputs a solution $S$ that satisfies a bicriteria approximation ratio of $(1/e - \epsilon, \frac{2}{\epsilon})$ in expectation in at most $O(\frac{nv}{\epsilon})$ number of queries.*

The analysis and the proof are deferred to Section B in the supplementary material. From the results, we can get that using `nonmono-bi` as a subroutine in `convert-rand` yields a $(\frac{2(1+\alpha)}{\epsilon}, 1/e - 2\epsilon)$-bicriteria approximation algorithm for nonmonotone SCP. The algorithm runs in $O(\frac{n|OPT|\log_{1+\alpha}(|OPT|)\ln(1/\delta)}{\epsilon\ln(1+\epsilon^2)})$ number of queries.

## 2.2 MONOTONE SUBMODULAR COVER WITH KNAPSACK PARTITION CONSTRAINTS

We now consider the problem of Monotone Submodular Cover with Knapsack Partition Constraints (SCKP). SCKP models the setting where we want to balance the cost across different partitions, and the costs of different elements are nonuniform for different elements in the ground set. We illustrate an example of SCKP. Consider a neural network training task in deep learning, we want to select pretraining data points from $N$ different predefined groups, and the goal is to select a subset of data points with minimal budget of the cost, where the cost of each data point may reflect computational, labeling, or storage expenses. Simultaneously, we would want a solution with balanced cost allocation across predefined groups in the dataset while ensuring that the submodular utility function (e.g., coverage of diverse features) meets a specified threshold $\tau$.

More formally, the definition of SCKP is as follows. Define a cost function $c : U \to \mathbb{R}_{\geq 0}$, and let $c(S) = \sum_{x \in S} c(x)$ for any subset $S \subseteq U$. Then SCKP is:

$$
\begin{aligned}
&\min v \\
&\text{s.t. } f(S) \geq \tau \\
&\quad c(S \cap U_j) \leq p_j v, \qquad \forall j \in [N].
\end{aligned}
\tag{1}
$$

In the definition of SCKP, $v$ represents the budget for the total cost. More specifically, $v$ is the upper bound on the total cost. The second constraint ensures that the cost of the solution set $S$ within each partition $U_j$ does not exceed a specified fraction, $p_j$, of $v$. Without loss of generality, we assume that $\sum_{j \in [N]} p_j = 1$.

This formulation naturally arises in various real-world applications, including influence maximization in social network analysis, where activating different nodes incurs varying costs, pretraining data selection for deep learning where different data points might require different computational or memory costs, and task allocation in multi-agent systems. Please see the Appendix C.1 for a detailed discussion on the motivating examples. In the following part, we discuss the pretraining data selection as a motivating example of the SCKP problem.

To address SCKP, we first propose an algorithm for the dual problem of Submodular Maximization with Knapsack Partition Constraint (SMKP), which is defined as $\arg\max\{f(S) : \sum_{s \in X \cap U_j} c(s) \leq p_j v, \forall j \in [N]\}$. Notice that when the cost is uniform, i.e., $c(s) = c, \forall s \in U$. SMKP is a monotone submodular maximization problem with a partition matroid constraint. Therefore, we can apply any submodular maximization algorithm with a matroid constraint. However, the output of standard algorithms can't achieve an objective value arbitrarily close to the optimal. Therefore, we propose the `greedy-knapsack-bi` algorithm, which proceeds in $\phi := \frac{\ln \frac{1}{\epsilon}}{\ln 2}$ blocks. The pseudocode of `greedy-knapsack-bi` is described in Algorithm 2. Within each block from Line 4 to Line 11, the algorithm visits each partition $U_j$ in the ground set $U$ and adds elements greedily with the highest density of marginal gain. Below we present the theoretical guarantee of `greedy-knapsack-bi` for SMKP.

**Theorem 2.5.** *Suppose that `greedy-knapsack-bi` described in Algorithm 2 is run for an instance of SMKP, then `greedy-knapsack-bi` outputs a solution set that satisfies*

$$
f(S) \geq (1 - \epsilon) f(OPT)
$$

$$
c(S \cap U_j) \leq \frac{2 \ln \frac{1}{\epsilon}}{\ln 2} p_j v, \qquad \forall j \in [N],
$$

*where $OPT$ is the optimal solution of SMKP.*

This theorem guarantees that the solution set returned by `greedy-knapsack-bi` achieves an objective value arbitrarily close to the optimal while the cost constraints are satisfied up to a violation factor of $\frac{2 \ln \frac{1}{\epsilon}}{\ln 2}$. In the special case of uniform costs, the theorem implies that `greedy-knapsack-bi` achieves a $(1 - \epsilon, O(\ln \frac{1}{\epsilon}))$ bicriteria approximation guarantee. This matches the best-known approximation ratio for monotone submodular maximization under a cardinality constraint. The improved performance stems from the blockwise structure of `greedy-knapsack-bi`, which effectively leverages the intrinsic similarity between the submodular maximization problem under the cardinality constraint and the partition matroid constraint. The proof and analysis of Theorem 2.5 are deferred to Appendix C.3.

### 2.2.1 CONVERTING THEOREM FOR SCKP

In order to convert any bicriteria algorithms for SMKP into bicriteria algorithms for SCKP, we propose and analyze the converting algorithm, denoted as `convert`. The pseudocode for `convert` is in Algorithm 5, and its theoretical guarantee is outlined in Theorem C.3, both in Section C.4 of the appendix. By leveraging the result in the converting theorem, we can obtain the following corollary.

---

**Algorithm 2** `greedy-knapsack-bi`

---

1: **Input:** $\epsilon$, an instance of SMKP
2: **Output:** solution set $S \subseteq U$
3: **for** $i = 1$ to $\frac{\ln \frac{1}{\epsilon}}{\ln 2}$ **do**
4:     **for** $j = 1$ to $N$ **do**
5:         $A \leftarrow \emptyset$, $B_j \leftarrow p_j v$.
6:         **while true do**
7:             $s \leftarrow \text{argmax}_{x \in U_j / S, c(x) \leq B_j} \frac{\Delta f(S \cup A, x)}{c(x)}$
8:             $A \leftarrow A \cup \{s\}$
9:             **if** $c(A) \geq B_j$ **then**
10:                 $S \leftarrow S \cup A$
11:                 **break**
12: **return** $S$

---

**Corollary 2.6.** *By using the* `greedy-knapsack-bi` *as a subroutine for the converting algorithm* `convert`*, we can obtain an algorithm for SCKP that returns a solution set $S$ and $v_S$ that satisfies*

$$v_S \leq \frac{2(1 + \alpha) \ln(1/\epsilon)}{\ln 2} v_{OPT}$$
$$f(S) \geq (1 - \epsilon)\tau$$
$$c(S \cap U_j) \leq p_j v_S$$

*where $OPT$ and $v_{OPT}$ are the optimal solution set and the optimal value of the problem SCKP defined in (1) respectively. The total runtime of the algorithm is upper bounded by $\mathcal{O}(n^2 \log_{1+\alpha}(\frac{c_{\max} n}{c_{\min}}))$. Here $c_{\max}$ and $c_{\min}$ are the maximum and minimum values of the cost of a single element respectively.*

## 2.3 SUBMODULAR COVER WITH FAIRNESS CONSTRAINT

In this section, we consider the monotone Submodular Cover problem with Fairness Constraint (SCF), recently proposed by Chen et al. (2025) (also referred to as Fair Submodular Cover in Chen et al. (2025)). SCF is defined as follows: Given a monotone and submodular function $f$, a threshold $\tau$, and bounds $p_c$ and $q_c$ on the proportion limits of the elements in each group, SCF aims to find

$$\text{argmin}_{S \in U} |S|$$
$$s.t. \quad p_c |S| \leq |S \cap U_c| \leq q_c |S|, \qquad \forall c \in [N]$$
$$f(S) \geq \tau.$$

Chen et al. proposed a converting approach that takes bicriteria algorithms for the dual problem of Submodular Maximization with Fairness constraint (SMF) (El Halabi et al., 2020) and converts them into algorithms for SCF. Formally, SMF seeks to maximize $f(S)$ subject to the constraint that $S \in \mathcal{M}_{fair}$. $\mathcal{M}_{fair}$ represents the fairness matroid and is defined as $\mathcal{M}_{fair} = \{S \subseteq U : |S \cap U_c| \leq u_c, \forall c \in [N], \sum_{c \in [N]} \max\{|S \cap U_c|, l_c\} \leq k\}$. If we set $l_c = 0$ for all $c \in [N]$ and we set $k = \sum_{c \in [N]} u_c$, then $\mathcal{M}_{fair}$ is equivalent to $\mathcal{M}_{fair} = \{S \subseteq U : |S \cap U_c| \leq u_c, \forall c \in [N]\}$. Therefore, SMF can be viewed as a generalized form of submodular maximization with a partition matroid constraint. A bicriteria approximation guarantee for SMF is defined as follows.

**Definition 2.7.** A discrete algorithm for SMF with an $(\alpha, \beta)$-bicriteria approximation ratio returns a solution $X$ such that

$$f(X) \geq \alpha f(OPT),$$
$$|X \cap U_c| \leq \beta u_c \qquad \forall c \in [N],$$
$$\sum_{c \in [N]} \max\{|X \cap U_c|, \beta l_c\} \leq \beta k.$$

Here $OPT$ is the optimal solution of the problem SMF, i.e., $OPT = \arg \max_{S \in \mathcal{M}_{fair}} f(S)$.

---

**Algorithm 3** `Block-Fair-Bi`

---

1: **Input:** fairness parameters
2: **Output:** $S \in U$
3: $S \leftarrow \emptyset$
4: **for** $i = 1$ **to** $\frac{\ln(1/\epsilon)}{\ln 2}$ **do**
5:      $B \leftarrow \emptyset$
6:      **while** $\exists s$ **s.t.** $B \cup \{s\} \in \mathcal{M}_{fair}$ **do**
7:          $x \leftarrow \arg\max_{s \in U, B \cup \{s\} \in \mathcal{M}_{fair}} \Delta f(S, s)$
8:          $B \leftarrow B \cup \{x\}$
9:          $S \leftarrow S \cup \{x\}$
     **return** $S$

---

Therefore, in order to leverage the conversion approach of Chen et al., we propose an algorithm for SMF called `Block-Fair-Bi` that uses a greedy block formation technique. `Block-Fair-Bi` is an improvement over the `greedy-fairness-bi` in Algorithm 5 in Chen et al. (2025) . This leads to a bicriteria approximation guarantee for SCF of $(\frac{1+\ln\frac{1}{\epsilon}}{\ln 2}, 1-\epsilon)$, which is a significant improvement compared to the best-known results for discrete algorithms of $(\frac{1}{\varepsilon}+1, 1-\mathcal{O}(\varepsilon))$ in Chen et al..

We now describe `Block-Fair-Bi`, pseudocode for which is presented in Algorithm 3. `Block-Fair-Bi` adopts a similar approach of running greedy algorithms in blocks. By definition, the $\beta$-extension of the fairness matroid constraint $\mathcal{M}_{fair}$ is given by $\mathcal{M}_{\beta} := \{S \subseteq U : |S \cap U_c| \leq \beta u_c, \forall c \in [N], \sum_{c \in [N]} \max\{|S \cap U_c|, \beta l_c\} \leq \beta\kappa\}$ as introduced in Chen et al. (2025). Therefore, an algorithm with a bicriteria approximation ratio of $(\alpha, \beta)$ for the SMF problem outputs a solution set that belongs to $\mathcal{M}_{\beta}$. The key intuition behind `Block-Fair-Bi` is that any set in the $\mathcal{M}_{\beta}$ can be expressed as the union of $\beta$ subsets, each belonging to $\mathcal{M}_{fair}$. This motivates our algorithm of dividing the capacity of the solution set into $\beta$ blocks. Here $\beta := \frac{\ln(1/\epsilon)}{\ln 2}$. Within each block from Line 5 to Line 9 in Algorithm 3, the algorithm operates in a greedy fashion: it iteratively adds the element with the highest marginal gain while maintaining $B \in \mathcal{M}_{fair}$. The main theoretical result is stated in Theorem 2.8 below. The proof of the theorem is deferred to Appendix D.

**Theorem 2.8.** *Suppose that `Block-Fair-Bi` is run for an instance of SMF, then `Block-Fair-Bi` outputs a solution $S$ that satisfies a $(1-\varepsilon, \frac{\ln\frac{1}{\varepsilon}}{\ln 2})$-bicriteria approximation guarantee in at most $O\left(n\kappa\ln(1/\varepsilon)\right)$ queries of $f$.*

## 3 EXPERIMENTS

In this section, we present an empirical evaluation of our proposed algorithms. In particular, we evaluate our `nonmono-bi` algorithm on instances of graph cut in Section 3.1. Next, we evaluate `greedy-knapsack-bi` on set cover instances and `Block-Fair-Bi` on both max cover and image summarization tasks in Sections 3.2 and 3.3, respectively. Additional details about the applications, setup, and results can be found in Section F in the supplementary material.

### 3.1 EXPERIMENTS ON NONMONOTONE SCP

We evaluate the performance of our algorithms on several instances of graph cut over social network data. The dataset used in the main paper is the email-EuAll dataset (n = 265214, 420045 edges) from the SNAP large network collection (Leskovec and Sosič, 2016). We compare the solutions returned by the `convert-rand` algorithm with four subroutines including: (i). our `nonmono-bi` algorithm ("BLOCK-G") (ii). the streaming algorithm from Feldman et al. (2018) ("STREAM"); (iii). the randomized algorithm of Chen et al. (2024b), initialized with the twin-greedy solution proposed in Han et al. (2020) ("GUIDED-RG"). (iv). The random-greedy algorithm for the submodular maximization problem with the cardinality constraint being the input guess of budget $\kappa$ (Buchbinder et al., 2014) ("RG"). The algorithms are evaluated in terms of the function value $f(S)$ returned by the solution $S$, the query complexity, and the minimum value of budget $v$ that satisfies the partition constraints, i.e., $\max_{i \in [N]} \frac{|S \cap U_i|}{p_i}$, and the execution time. The results are compared for different values of the threshold $\tau$.

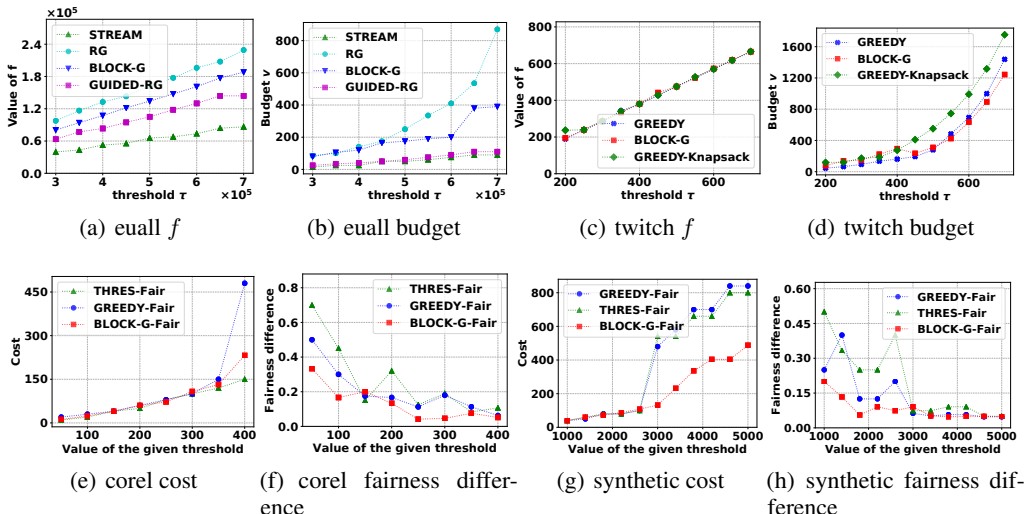

(a) euall $f$      (b) euall budget      (c) twitch $f$      (d) twitch budget

(e) corel cost      (f) corel fairness difference      (g) synthetic cost      (h) synthetic fairness difference

Figure 1: The experimental results of running the algorithms on the euall dataset, the twitch dataset, the Corel5k dataset, and the synthetic dataset. Budget: $\max_{i \in [N]} \frac{c(S \cap U_i)}{p_i}$. Cost: the size of the solution. Fairness difference: $(\max_c |S \cap U_c| - \min_c |S \cap U_c|)/|S|$.

The results in terms of $f$ and the minimum budget are described in Figure 1(a) and 1(b). From the results, one can see that RG and BLOCK-G consistently achieve higher objective values $f$ than the other methods. This is consistent with our theoretical results as the approximation ratio on the function value of these two algorithms satisfies $f(S) \geq (1/e - \epsilon)\tau$ and $f(S) \geq (1/e - 2\epsilon)\tau$ respectively while the other two algorithms STREAM, GUIDED-RG achieve worse approximation ratios on the function value of $1/0.583 - \epsilon$, $0.305 - 2\epsilon$ respectively. However, in terms of budget, we can see that the RG algorithm performs poorly, since it does not account for partition constraints, resulting in imbalanced budget allocations. The STREAM and GUIDED-RG algorithm returns solutions with smaller budgets since both these two algorithms achieve a bicriteria approximation ratio such that $v_S \leq (1 + \alpha)v_{OPT}$. While BLOCK-G has a higher budget due to its weaker bicriteria guarantee of $v_S \leq \frac{2(1+\alpha)}{\epsilon} v_{\text{OPT}}$, it does achieve a significantly higher function value.

### 3.2 EXPERIMENTS ON SCKP

We evaluate three different algorithms on the instance of max cover: the converting algorithm `convert` with two different subroutines: `greedy-knapsack-bi` ("BLOCK-G") and a standard greedy algorithm without the block-wise structure ("GREEDY"), and the greedy algorithm for submodular cover without the partition-knapsack constraint ("GREEDY-Knapsack"). Further details regarding the GREEDY and the GREEDY-Knapsack algorithms, and the experimental setup are provided in Appendix F.1.

The results in terms of the minimum budget, which can be calculated by $\max_{i \in [N]} \frac{c(S \cap U_i)}{p_i}$, and the function value $f$ are plotted in Figure 1(c) and Figure 1(d). From the results, one can see that the $f$ values of solutions returned by BLOCK-G, GREEDY-Knapsack, and GREEDY are nearly the same. This is because the theoretical guarantees on $f$ are about the same for the different algorithms. However, the budget of the solution returned by our algorithm BLOCK-G is smaller than the other two algorithms, which demonstrates the effectiveness of our approach of running greedy in blocks.

### 3.3 EXPERIMENTS ON SCF

For the SCF problem, we evaluate algorithms using the conversion framework from Chen et al. (2025) with different subroutines: our `Block-Fair-Bi` algorithm ("BLOCK-G-Fair"), the standard greedy algorithm ("GREEDY-Fair") and the threshold greedy algorithm ("THRES-Fair"). These algorithms are compared in terms of solution cost (cardinality), fairness difference, objective function value,

query complexity, and execution time for varying values of the threshold $\tau$. Here we set $\alpha = 0.2$ for the converting algorithms in Algorithm 1 in Chen et al. (2025). The parameters in the fairness constraint are set to be $u_c = 1.1/N, l_c = 0.9/N$. (where $N$ is the number of groups). Additional details about the applications, setup, and results can be found in Section F in the appendix.

Figures 1(e) and 1(g) illustrate the cost (cardinality) of the solution sets, while Figures 1(f) and 1(h) show the fairness differences across varying $\tau$ values. In most cases, BLOCK-G-Fair achieves a lower cost than THRES-Fair and GREEDY-Fair, aligning with our theoretical results. In the case where $\tau$ is large on the corel dataset, the cost of the THRES-Fair is smaller than the BLOCK-G-Fair. However, Figures 1(f) and 1(h) reveal that fairness differences in THRES-Fair and GREEDY-Fair are significantly larger than in BLOCK-G-Fair, demonstrating that BLOCK-G-Fair produces more balanced solutions. This is expected given that the fairness constraint from Chen et al. (2025) ensures that $\beta \lfloor \frac{p_c|S|}{\beta} \rfloor \leq |S \cap U_c| \leq \beta \lceil \frac{q_c|S|}{\beta} \rceil$, which means the solution set might break the fairness constraint by an additive factor of $\beta$. Notably, $\beta = \mathcal{O}(1/\epsilon)$ for the THRES-Fair and GREEDY-Fair and $\beta = \frac{\ln(\frac{1}{\epsilon})}{\ln 2}$ for BLOCK-G-Fair, which means our method achieves an enhanced fairness guarantee.

## 4 REPRODUCIBILITY STATEMENT

All theoretical results in this paper are supported by complete proofs, which are provided in the main text and the appendix. Approximation guarantees and detailed proofs are described to ensure that the theoretical contributions can be independently verified.

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

---

**Algorithm 4** `convert-rand`

---

**Input**: An SCP instance with threshold $\tau$, a $(\gamma, \beta)$-bicriteria approximation algorithm for SMP, $\alpha > 0$

**Output**: $S \subseteq U$

1: $S_i \leftarrow \emptyset, \forall i \in \{1, ..., \ln(1/\delta)/\ln(\frac{\beta-\gamma+\epsilon}{\beta-\gamma})\}$
2: $g \leftarrow (1 + \alpha)$
3: **while** $f(S_i) < (\gamma - \epsilon)\tau \; \forall i$ **do**
4:     **for** $i \in \{1, ..., \ln(1/\delta)/\ln(\frac{\beta-\gamma+\epsilon}{\beta-\gamma})\}$ **do**
5:         $S_i \leftarrow (\gamma, \beta)$-bicriteria approximation for SMP with objective function $f$ and budget $g$
6:     $g \leftarrow (1 + \alpha)g$
7: **return** $S$

---

| Problem | Algorithm | Approximation Ratio | Query Complexity |
|---------|-----------|---------------------|------------------|
| Nonmonotone SCP | `convert-rand` + `nonmono-bi` | $\left(O\left(\frac{1+\alpha}{\epsilon}\right), \frac{1}{e} - \epsilon\right)$ | $\mathcal{O}\left(n^2 \log_{1+\alpha} n \ln(1/\delta)/\epsilon/\ln(1+\epsilon)\right)$ |
| Monotone SCKP | `greedy-knapsack-bi` + `convert` | $\left(\frac{(1+\alpha)\ln(1/\epsilon)}{\ln 2}, 1 - \epsilon\right)$ | $\mathcal{O}\left(n^2 \log_{1+\alpha}\left(\frac{c_{\max}n}{c_{\min}}\right)\right)$ |
| Monotone SCF | `Block-Fair-Bi` + `convert-fair` (Chen et al., 2025) | $(\mathcal{O}(\ln(1/\epsilon)), 1 - \epsilon)$ | $\mathcal{O}\left(\frac{n \log n\kappa \ln(1/\delta)}{\epsilon}\right)$ |

Table 1: Summarization of approximation algorithms for Submodular Cover Problems in this paper.

## APPENDIX

## A THE USE OF LARGE LANGUAGE MODELS (LLMS)

In this paper, we use Large Language Models solely to polish the writing to improve the clarity and presentation. All theoretical results and technical contributions in this paper were not developed by LLMs.

## B APPENDIX FOR SECTION 2.1

In this section, we present missing content from Section 2.1, where we considered non-monotone submodular cover with partition constraints. First, pseudocode for the converting algorithm `convert-rand`, which we only informally described in Section 2.1, is given in Algorithm 4. Next, we present the omitted proofs of Theorems 2.3 and Theorem 2.4.

**Theorem 2.3.** *Any randomized $(\gamma, \beta)$-bicriteria approximation algorithm for SMP that runs in time $\mathcal{T}(n)$ where $\gamma$ holds only in expectation can be converted into a $((1 + \alpha)\beta, \gamma - \epsilon)$-bicriteria approximation algorithm for SCP that runs in time $O(\log_{1+\alpha}(|OPT|) \ln(1/\delta)\mathcal{T}(n))/\ln(\frac{\beta-\gamma+\epsilon}{\beta-\gamma}))$ where $\gamma$ holds with probability at least $1 - \delta$.*

*Proof.* Consider the run of the algorithm for SMP on Line 5 of Algorithm 4 when the guess of optimal value $g$ falls into the region

$$v_{OPT} \leq g \leq (1 + \alpha)v_{OPT}.$$

Let us denote the partition matroid with budget $g$ as $\mathcal{M}$, i.e., $\mathcal{M} := \{S \subseteq U : |S \cap U_j| \leq p_j v, \forall j \in [N]\}$. The SMP problem is then defined to find $\arg\max\{f(S) : S \in \mathcal{M}\}$. We denote the optimal solution of the SMP problem with budget $g$ as $OPT_g$, i.e.,

$$OPT_g := \arg\max\{f(S) : |S \cap U_j| \leq p_j v, \forall j \in [N]\}.$$

Besides, we define the optimal solution for SCP as $OPT$. By the fact that the optimal solution $OPT$ is feasible for SCP, we have that

$$|OPT \cap U_j| \leq p_j v_{OPT} \leq p_j v \tag{2}$$

which means that $OPT \in \mathcal{M}$, and therefore $f(OPT) \leq \max_{S \in \mathcal{M}} f(S) = f(OPT_g)$. Since $f(OPT) \geq \tau$, we have $f(OPT_g) \geq \tau$. It then follows that for each $i \in \{1, ..., \ln(1/\delta)/\ln(\frac{\beta-\gamma+\epsilon}{\beta-\gamma})\}$,

$$P\Big(f(S_i) \leq (\gamma - \epsilon)\tau\Big) \leq P\Big(f(S_i) \leq (\gamma - \epsilon)f(OPT_g)\Big)$$
$$\leq P\Big(\beta f(OPT_g) - f(S_i) \geq (\beta - \gamma + \epsilon)f(OPT_g)\Big)$$

By the theoretical guarantees of the algorithm for SMP, we have that for all $i \in \{1, ..., \ln(1/\delta)/\ln(\frac{\beta-\gamma+\epsilon}{\beta-\gamma})\}$, we have that $\mathbb{E}f(S_i) \geq \gamma f(OPT_g)$ and $|S_i \cap U_j| \leq p_j \beta g$ for each $j \in [N]$. It then follows that

$$P\Big(f(S_i) \leq (\gamma - \epsilon)\tau\Big) \leq P\Big(\beta f(OPT_g) - f(S_i) \geq \frac{\beta - \gamma + \epsilon}{\beta - \gamma}(\beta f(OPT_g) - \mathbb{E}f(S_i))\Big)$$

Let us denote $\mathcal{M}_\beta := \{S \subseteq U : |S \cap U_j| \leq p_j \beta g, \forall j \in [N]\}$, i.e., $\mathcal{M}_\beta$ is the $\beta$-extension of the matroid $\mathcal{M}$ as is defined in Chen et al. (2025). Since $S_i$ satisfies $|S_i \cap U_j| \leq p_j \beta g$ for any $j \in [N]$, we have that $S_i \in \mathcal{M}_\beta$ for each $i \in \{1, ..., \ln(1/\delta)/\ln(\frac{\beta-\gamma+\epsilon}{\beta-\gamma})\}$. Notice that for any set $A \in \mathcal{M}_\beta$, we can express $A$ as the union of $\beta$ disjoint subsets in $\mathcal{M}$. Let us denote them as $A_1, A_2, ..A_m$. Thus we have that

$$f(A) = f(\cup_{i \in [\beta]} A_i)$$
$$\leq \sum_{i \in [\beta]} f(A_i) \leq \beta f(OPT_g),$$

where the first inequality follows from the submodularity of $f$. It then follows that $\max_{S \in \mathcal{M}_\beta} f(S) \leq \beta f(OPT_g)$. Notice that $S_i \in \mathcal{M}_\beta$, we have that $\beta f(OPT_g) - f(S_i) \geq 0$. Thus, we can apply Markov's inequality on the random variable $\beta f(OPT_g) - f(S_i)$. Therefore, we can get that for each $i \in \{1, ..., \ln(1/\delta)/\ln(\frac{\beta-\gamma+\epsilon}{\beta-\gamma})\}$

$$P\Big(f(S_i) \leq (\gamma - \epsilon)\tau\Big) \leq P\Big(\beta f(OPT_g) - f(S_i) \geq \frac{\beta - \gamma + \epsilon}{\beta - \gamma}(\beta f(OPT_g) - \mathbb{E}f(S_i))\Big)$$
$$\leq \frac{\beta - \gamma}{\beta - \gamma + \epsilon}$$

Then the probability that none of the subsets $S_i$ can reach the stopping condition can be bounded by

$$P(f(S_i) \leq (\gamma - \epsilon)\tau, \forall i) = P(f(S_i) \leq (\gamma - \epsilon)\tau, \forall i)$$
$$= \prod_{i=1}^{\ln(1/\delta)/\ln(\frac{\beta-\gamma+\epsilon}{\beta-\gamma})} P(f(S_i) \leq (\gamma - \epsilon)\tau)$$
$$\leq (\frac{\beta - \gamma}{1 + \epsilon - \gamma})^{\ln(1/\delta)/\ln(\frac{\beta-\gamma+\epsilon}{\beta-\gamma})} = \delta.$$

This means with probability at least $1 - \delta$, `convert-rand` stops when $g$ reaches the region where $v_{OPT} \leq g \leq (1 + \alpha)v_{OPT}$ since the condition of the **while** loop is not satisfied. Therefore, by the assumption that the subroutine algorithm is a $(\gamma, \beta)$-bicriteria approximation algorithm, we have that the output solution $S$ satisfies that $|S \cap U_j| \leq p_j \beta g \leq p_j \beta (1 + \alpha)v_{OPT}$. Then the objective value of the optimal solution $S$ can be set to be $v_S = \beta(1 + \alpha)v_{OPT}$. It also implies that there are at most $O(\log_{1+\alpha} v_{OPT})$ number of guesses of the cardinality of the optimal solution. Since for each guess, we run the SMP for $\ln(1/\delta)/\ln(\frac{\beta-\gamma+\epsilon}{\beta-\gamma})$ times, the algorithm runs in time $O(\log_{1+\alpha}(v_{OPT}) \ln(1/\delta)\mathcal{T}(n)/\ln(\frac{\beta-\gamma+\epsilon}{\beta-\gamma}))$.

$\square$

Next, we present the proof for the result in Theorem 2.4.

**Theorem 2.4.** *Suppose that* `nonmono-bi` *is run for an instance of nonmonotone SMP with budget $v$, then* `nonmono-bi` *outputs a solution $S$ that satisfies a bicriteria approximation ratio of $(1/e - \epsilon, \frac{2}{\epsilon})$ in expectation in at most $O(\frac{nv}{\epsilon})$ number of queries.*

*Proof.* Let us denote the solution set after adding the $l$-th element in $j$-th subgroup during the $i$-th round of the outer for loop in Line 6 in Algorithm 1 as $S_{i,j,l}$, and we define the solution set after completing adding all the elements in the $j$-th subgroup during the $i$-th round in Algorithm 1 as $S_{i,j}$. For notation simplicity, we also define $\phi := \frac{2}{\epsilon}$. From the greedy selection strategy, we have that

$$\mathbb{E}[f(S_{i,j,l}) - f(S_{i,j,l-1})] \geq \frac{\sum_{a \in OPT \cap U_j} \Delta f(S_{i,j,l-1}, a)}{p_j v \phi}.$$

By submodularity, we would have that

$$\mathbb{E}[f(S_{i,j,l}) - f(S_{i,j,l-1})] \geq \frac{\sum_{a \in OPT \cap U_j} \Delta f(S_{i,j}, a)}{p_j v \phi}.$$

By summing over all $l \in [p_j v]$, it then follows that

$$\mathbb{E}[f(S_{i,j}) - f(S_{i,j-1})] \geq \frac{\sum_{a \in OPT \cap U_j} \Delta f(S_{i,j}, a)}{\phi}.$$

Let us denote the solution after completing the entire $i$-th round as $S_i$. By submodularity, it then follows that

$$\mathbb{E}[f(S_{i,j}) - f(S_{i,j-1})] \geq \frac{\mathbb{E}[\Delta f(S_{i,j}, OPT_j)]}{\phi}$$
$$\geq \frac{\mathbb{E}[\Delta f(S_i, OPT_j)]}{\phi},$$

By summing over all $j \in [N]$, we have

$$\mathbb{E}[f(S_i) - f(S_{i-1})] \geq \frac{\sum_{j=1}^{N} \mathbb{E}[\Delta f(S_i, OPT_j)]}{\phi}$$
$$\geq \frac{\mathbb{E}[\Delta f(S_i, OPT)]}{\phi}.$$

Then it follows that

$$\mathbb{E}[f(S_i) - f(S_{i-1})] \geq \mathbb{E}[\frac{f(S_i \cup OPT) - f(S_i)}{\phi}].$$

Notice that by the greedy selection step, for each group $j$ and each element $a \in OPT \cap U_j$ appears in $S_i$ with probability at most $1 - (1 - \frac{1}{p_j v \phi})^{p_j v i}$. Since $(1 - \frac{1}{x})^x$ increases with $x$ in the range of $[1, +\infty)$, we have that $(1 - \frac{1}{p_j v \phi})^{p_j v \phi} \geq (1 - \frac{1}{\phi})^{\phi}$. Therefore, we would get $1 - (1 - \frac{1}{p_j v \phi})^{p_j v i} \leq 1 - (1 - \frac{1}{\phi})^i$. From Lemma 2.2 in Buchbinder et al. (2014), we can conclude that

$$\mathbb{E}[f(S_i \cup OPT)] \geq (1 - \frac{1}{\phi})^i f(OPT).$$

By rearranging the above inequality, we can get that

$$\mathbb{E}[f(S_i)] \geq \frac{\phi}{\phi + 1} \mathbb{E}[f(S_{i-1})] + \frac{1}{\phi + 1}(1 - \frac{1}{\phi})^i f(OPT).$$

By induction, we have that the output solution set satisfies that

$$\mathbb{E}[f(S)] = \mathbb{E}[f(S_\phi)]$$

$$\geq \frac{1}{\phi+1} \sum_{i=1}^{\phi} (\frac{\phi}{\phi+1})^{\phi-i} (1 - \frac{1}{\phi})^i f(OPT)$$

$$\geq \frac{1}{\phi+1} \sum_{i=1}^{\phi} (\frac{\phi-1}{\phi})^{\phi-i} (1 - \frac{1}{\phi})^i f(OPT)$$

$$\geq \frac{\phi}{\phi+1} (1 - \frac{1}{\phi})^\phi f(OPT) \geq \frac{1}{e}(1 - \epsilon) f(OPT). \tag{3}$$

where the last inequality follows from the fact that $(1 - \frac{1}{\phi})^{\phi-1} \geq e^{-1}$ for any $\phi > 1$, and that $\phi = \frac{2}{\epsilon}$. $\qquad\square$

## C  APPENDIX FOR SECTION 2.2

We now present omitted content from Section 2.2, where we studied monotone submodular cover with knapsack partition constraints. Our first goal is to provide additional detail to motivate and explain our proposed optimization problem formulation. In particular, several detailed motivating examples of SCKP are presented in Section C.1, and then further we provide detailed discussion on the formulation of SCKP in Section C.2. Then in Section 2.2, we provide the omitted proofs from Section 2.2 in the main paper. Namely, we present the missing proofs of the theoretical guarantee for the `Block-Greedy` algorithm with `Alg-SM` as the subroutine, stated in Theorem 2.5, and we present the converting algorithm `convert` for transforming an algorithm for SMKP to an algorithm for SCKP, stated in Theorem C.3.

### C.1  MOTIVATING APPLICATIONS OF SCKP

In this portion of the appendix, we provide a series of examples to motivate our study of the SCKP problem, where the objective is to find a solution set $S$ which minimizes the total cost while maintaining a certain level of utility ($f(S) \geq \tau$) and a balanced cost constraint across different partitions ($c(S \cap U_j) \leq p_j v$). The motivating examples of this problem include

- **Influence Maximization**: In this application, we might want to select a set of nodes with minimum cost (e.g., limited budget funds to be allocated) while ensuring a certain level of influence spread. The cost should also be balanced among each partition of the universe, which is splitted by the demographic or geographic attributes. Different nodes (e.g., influential users or groups) may require different costs to be activated (e.g., through targeted ads or promotions), and thus the cost is non-uniform among different nodes.

- **Pretraining Data Selection**: In pretraining data selection, the goal is to select a subset of data points with minimal cost, where costs may reflect computational, labeling, or storage expenses. The problem involves balancing costs across predefined groups in the dataset while ensuring that the utility function (e.g., coverage of diverse features) meets a specified threshold.

- **Multi-Agent Task Allocation:** The objective is to find a set of tasks that minimizes the total cost of the assigned tasks while achieving an overall utility or performance of a certain level and a balanced cost across different types of tasks (e.g., delivery, inspection, or cleaning). Tasks have different execution costs depending on complexity, duration, or required resources and thus the cost is nonuniform.

### C.2  CLARIFICATION OF THE PROBLEM DEFINITION OF SCKP

In this section, we provide some illustrations of the problem formulation of SCKP defined in Section 2.2 in the main paper. First of all, recall that the classical Minimum Cost Submodular Cover (MCSC) studied in previous work (Iyer and Bilmes, 2013; Crawford, 2019) is defined as $\arg\min\{c(S) : f(S) \geq \tau\}$ where $c : 2^U \to \mathbb{R}$ is a modular, positive cost function. In our setting,

we also want to ensure a balanced budget allocation across different partitions. Therefore, one of the definition of our problem should be $\arg\min\{c(S) : f(S) \geq \tau, c(S \cap U_j) \leq p_j c(S), \forall j \in [N]\}$.

However, the problem defined above can have feasibility issues in many cases. In particular, the constraint of $c(S \cap U_j) \leq p_j c(S)$ for each $j \in [N]$ can be really hard to satisfy, and can even render the problem infeasible. For example, if we set $p_j = 1/N$ for each $j \in [N]$, and that for each $s \in U_{j_1}$ $c(s) = \pi$, and each $s \in U_{j_2}$, $c(s) = 1$. From the definition of $(P1)$, we can see that there is no subset $S \subseteq U$ that satisfies $c(S \cap U_j) \leq p_j c(S)$ for each $j \in [N]$.

To solve this feasibility issue, we can relax the constraint on the balanced solution such that it can be slightly broken by the cost of a single element. Let us define $c_j = \max\{c(s) : s \in U_j\}$ to be the maximum singleton cost within the partition $U_j$. It then follows that the definition of the relaxed problem should be $\arg\min\{c(S) : f(S) \geq \tau, c(S \cap U_j) \leq p_j c(S) + c_j, \forall j \in [N]\}$

For notation simplicity, we use $(P1)$ to denote this problem, i.e.,

$$(P1): \quad \min_{S \subseteq U} c(S)$$
$$f(S) \geq \tau$$
$$c(S \cap U_j) \leq p_j c(S) + c_j, \qquad \forall j \in [N]. \tag{4}$$

To solve this problem, we can slightly relax the constraint on the cost by introducing another variable $\mu$ to the constraint, i.e., $c(S \cap U_j) \leq p_j \mu c(S)$ for each $j \in [N]$. Notice that here we also want to minimize the level of breaking the constraint, to do that, we replace the objective function from minimizing $c(S)$ to $\mu c(S)$ in the optimization problem, Next, by replacing the term $\mu c(S)$ with $v$, we obtain the definition of the SCKP problem. Additionally, compared with the optimization problem defined in $(P1)$, the problem defined in SCKP in (1) preserves the feasibility as long as the threshold $\tau$ satisfies $f(U) \geq \tau$.

$$(P2): \quad \min_{S \subseteq U} v(S)$$
$$f(S) \geq \tau$$
$$c(S \cap U_j) \leq p_j v, \qquad \forall j \in [N], \tag{5}$$

Let us define the optimal solution and optimal value of $(P1)$ as $OPT_{P1}$ and $v_{OPT}$ respectively, and we denote the optimal solution of SCKP defined as $OPT$. It is worth noting that the optimal solution in the optimization problem $(P1)$ has a similar quality to our SCKP problem $(P2)$. In particular, we have that the optimal value of $P1$ and $(P2)$ satisfies the following lemma:

**Lemma C.1.** *The optimal value of $(P1)$ and $(P2)$ satisfies*

1. *$v_{OPT} \leq c(OPT_1) + \max_{i \in [N]} \frac{c_i}{p_i}$.*

2. *When the optimal value of $(P2)$ satisfies that $p_j v_{OPT} \leq c(U_j)$, we have that $c(OPT_1) \leq \alpha v_{OPT} + \sum_{i \in [N]} c_j$, where $\alpha = \sum_{j \in [N]} p_j$.*

*Proof.* We first prove the first part of the lemma. Since $OPT_1$ is feasible for problem $(P1)$, it must satisfy all constraints of $(P1)$. In particular, for each $j \in [N]$, we have:

$$c(OPT_1 \cap U_j) \leq p_j c(OPT_1) + c_j$$
$$= p_j \left( c(OPT_1) + \frac{c_j}{p_j} \right)$$
$$\leq p_j \left( c(OPT_1) + \max_{i \in [N]} \frac{c_i}{p_i} \right) \tag{6}$$

Setting $v = c(OPT_1) + \max_{i \in [N]} \frac{c_i}{p_i}$, we observe that $S = OPT_1$ satisfies the constraints of the SCKP problem defined in $(P2)$. Therefore, $v_{OPT} \leq v$, proving the first result. We prove the second result by constructing a set $A$ by the following procedure.

1. Initialize $A \leftarrow OPT_2$.

2. For $j = 1$ to $N$ do:

    (a) While $c(A \cap U_j) \leq p_j v_{\text{OPT}}$:
        i. $x \leftarrow \arg\min_{x' \in U_j \backslash A} c(x')$
        ii. $A \leftarrow A \cup \{x\}$

Notice that for each $j \in [N]$, $c(OPT_2 \cap U_j) \leq p_j v_{OPT}$, and that $c(U_j) \geq p_j v_{OPT}$. Therefore, upon the termination of the above procedure, set $A$ satisfies that

$$p_j v_{OPT} \leq c(A \cap U_j) \leq p_j v_{OPT} + c_j \qquad (7)$$

It then follows that $\sum_{j \in [N]} p_j v_{OPT} \leq \sum_{j \in [N]} c(A \cap U_j)$, and thus $\alpha v_{OPT} \leq c(A)$. Therefore, for each $j \in [N]$, we have that

$$c(A \cap U_j) \leq p_j v_{OPT} + c_j \leq p_j c(A) + c_j \qquad (8)$$

It implies that $A$ is feasible for problem $(P1)$, therefore, we can conclude $c(OPT_1) \leq c(A) \leq \sum_{j \in [N]} p_j v_{OPT} + c_j \leq \alpha v_{OPT} + \sum_{j \in [N]} c_j$. $\qquad \square$

Besides, we want to point out that another benefit of the definition of our problem is that it preserves the dual relationship between the SCKP problem and the SMKP problem, which is defined as $\arg\max f(S) : \sum_{s \in X \cap U_j} c(s) \leq p_j v$. In particular, here the variable $v$ in SMKP also serves as the budget of the cost constraint. This property facilitates our application of converting theorems, which is used to convert bicriteria algorithms for SMF to algorithms for SCF.

### C.3 PROOF OF THEOREM 2.5

In this portion of the appendix, we present the missing proofs of the theoretical guarantee for the `Block-Greedy` algorithm with `Alg-SM` as the subroutine. The theorem statement is provided in Theorem 2.5.

**Theorem 2.5.** *Suppose that* `greedy-knapsack-bi` *described in Algorithm 2 is run for an instance of SMKP, then* `greedy-knapsack-bi` *outputs a solution set that satisfies*

$$f(S) \geq (1 - \epsilon)f(OPT)$$

$$c(S \cap U_j) \leq \frac{2 \ln \frac{1}{\epsilon}}{\ln 2} p_j v, \qquad \forall j \in [N],$$

*where $OPT$ is the optimal solution of SMKP.*

Let us denote the solution set after adding the $l$-th element in $j$-th subgroup during the $i$-th round of the outer for loop from Line 4 to Line 11 in Algorithm 2 as $S_{i,j,l}$, and we define the solution set after completing adding all the elements in the $j$-th subgroup during the $i$-th round in in Algorithm 2 as $S_{i,j}$. Before we prove Theorem 2.5, we prove the result in the following lemma.

**Lemma C.2.** *Let $S_{i,j}$ be the solution set of the algorithm* `greedy-knapsack-bi` *in Algorithm 2 after completing adding all the elements in the $j$-th subgroup during the $i$-th round in in Algorithm 2, then we would get that*

$$f(S_{i,j}) - f(S_{i,j-1}) \geq \Delta f(S_{i,j}, OPT_j)$$

*where $OPT_j := OPT \cap U_j$ is the intersection of the optimal solution set $OPT$ and the $j$-th partition $U_j$, and that*

$$B_j \leq c(S_{i,j}/S_{i,j-1}) \leq 2B_j.$$

*Here $B_j := p_j v$.*

*Proof.* Let $A_{i,j,l}$ be the set $A$ after adding the $l$-th element to the subgroup $j$ in the iteration $i$, and let $s_{i,j,l}$ be the $l$-th element $s$ added to the set $A$ during the $i$-th outer loop in the subgroup $j$ in Algorithm 2. It then follows that for any element $o \in OPT_j$,

$$\frac{\Delta f(S_{i,j-1} \cup A_{i,j,l-1}, s_{i,j,l})}{c(s_{i,j,l})} \geq \frac{\Delta f(S_{i,j-1} \cup A_{i,j,l-1}, o)}{c(o)}.$$

By rearranging the above inequality, we can get

$$c(o)\Delta f(S_{i,j-1} \cup A_{i,j,l-1}, s_{i,j,l}) \geq c(s_{i,j,l})\Delta f(S_{i,j-1} \cup A_{i,j,l-1}, o).$$

Summing over all $o \in OPT_j$ and by submodularity, we can get

$$c(OPT_j)\Delta f(S_{i,j-1} \cup A_{i,j,l-1}, s_{i,j,l}) \geq c(s_{i,j,l})\Delta f(S_{i,j-1} \cup A_{i,j,l-1}, OPT_j)$$

Let us denote the total number of iterations in Algorithm 2 as $T$. By submodularity, it then follows that

$$c(OPT_j)\Delta f(S_{i,j-1} \cup A_{i,j,l-1}, s_{i,j,l}) \geq c(s_{i,j,l})\Delta f(S_{i,j-1} \cup A_{i,j,T}, OPT_j).$$

Since

$$\Delta f(S_{i,j-1} \cup A_{i,j,l-1}, s_{i,j,l}) = f(S_{i,j-1} \cup A_{i,j,l}) - f(S_{i,j-1} \cup A_{i,j,l-1}),$$

we can sum over all $l \in [T]$ and get

$$c(OPT_j)\{f(S_{i,j-1} \cup A_{i,j,T}) - f(S)\} \geq c(A_{i,j,T})\Delta f(S_{i,j-1} \cup A_{i,j,T}, OPT_j).$$

Since $S_{i,j-1} \cup A_{i,j,T} = S_{i,j}$, we have

$$c(OPT_j)\{f(S_{i,j}) - f(S)\} \geq c(A_{i,j,T})\Delta f(S_{i,j}, OPT_j).$$

By the stopping condition of Algorithm 2, we have that $c(A_{i,j,T-1}) \leq B_j$, therefore,

$$B_j \leq c(A_{i,j,T}) \leq 2B_j$$

Since $c(OPT_j) \leq B_j$, it then follows that

$$f(S_{i,j}) - f(S) \geq \Delta f(S_{i,j}, OPT_j).$$

We can then conclude the proof by the fact that $A_{i,j,T} = S_{i,j}/S_{i,j-1}$. $\square$

With Lemma C.2, we can prove the result of Theorem 2.5 as follows.

*Proof.* Let us denote the solution set after completing the $i$-th round in Algorithm 2 as $S_i$, i.e., $S_i = S_{i,N}$. By the result in Lemma C.2, it then follows that

$$f(S_{i,j}) - f(S_{i,j-1}) \geq \Delta f(S_{i,j}, OPT_j) \geq \Delta f(S_i, OPT_j),$$

where the second inequality follows from submodularity. Summing over all $j \in [N]$, we would get

$$f(S_i) - f(S_{i-1}) \geq \sum_{j \in [N]} \Delta f(S_i, OPT_j) \geq \Delta f(S_i, OPT)$$

Therefore, $f(S_i) \geq \frac{f(S_{i-1}) + f(OPT)}{2}$. By induction, we have that the final output solution set $S$ satisfies

$$f(S) = f(S_\phi) \geq (1 - \epsilon)f(OPT).$$

Notice that $S = \cup_{i=1}^{\phi} S_i/S_{i-1}$. From Lemma C.2, we can get $c((S_i/S_{i-1}) \cap U_j) \leq 2p_j v$. Therefore $c(S \cap U_j) = \sum_{i=1}^{\phi} c((S_i/S_{i-1}) \cap U_j) \leq \frac{2\ln\frac{1}{\epsilon}}{\ln 2} p_j v$. $\square$

### C.4 THEORETICAL ANALYSIS OF ALGORITHM 5

In this portion of the appendix, we present the converting algorithm `convert` for transforming an algorithm for SMKP to an algorithm for SCKP. The pseudocode is described in Algorithm 5. The theoretical guarantee of `convert` is provided in Theorem C.3.

**Theorem C.3.** *Suppose that we have an algorithm `Alg-SM` for SMKP, and given budget $v$, `Alg-SM` is guaranteed to return a set $S$ such that $f(S) \geq \gamma f(OPT_{SM})$ and $c(S \cap U_j) \leq \beta p_j v$, in time $T(n)$, where $OPT_{SM}$ is the optimal solution of SMKP. Then the algorithm `convert` using `Alg-SM` as a subroutine returns a set $S$ and a value $v_S$ in time $\mathcal{O}(\log_{1+\alpha}(\frac{c_{\max} n}{c_{\min}})T(n))$ such that $v_S \leq \beta(1 + \alpha)v_{OPT}$, $c(S \cap U_j) \leq p_j v_S$ and $f(S) \geq \gamma\tau$. Here $v_{OPT}$ is the optimal value of SCKP. $c_{\max}$ and $c_{\min}$ are the maximum and minimum values of the cost of a single element respectively.*

---

**Algorithm 5** `convert`

---

**Input**: $\alpha, \epsilon$
**Output**: $S \subseteq U$

1: $v_g \leftarrow (1+\alpha)c_{\min}, S \leftarrow \emptyset$
2: **while** $f(S) < \gamma\tau$ **do**
3:     $S \leftarrow$ Algorithm for SMKP run with budget parameter $v = v_g$
4:     $v_g \leftarrow (1+\alpha)v_g$
5: $v_S = \beta v_g$
6: **return** $S, v_S$

---

*Proof.* Let $OPT$ be the optimal solution to the instance of SCKP. Consider the iteration of `convert` where $v_g$ has just increased above $v_{OPT}$, i.e., $v_{OPT} \leq v_g \leq (1+\alpha)v_{OPT}$. Then we run `Alg-SM` with budget $v_{OPT} \leq v_g \leq (1+\alpha)v_{OPT}$. Then by the assumptions on `Alg-SM` we have that

$$f(S) \geq \gamma f(OPT_{SM}). \tag{9}$$

Notice that the optimal solution $OPT$ for SCKP satisfies that $c(OPT \cap U_j) \leq p_j v_{OPT} \leq p_j v_g$. It then follows that $OPT$ is feasible for the SMKP problem with input $v_g$. Let us denote the optimal solution of SMKP as $OPT_S M$. Then we have that

$$f(OPT_{SM}) \geq f(OPT).$$

Since $OPT$ is the optimal solution for SCKP, then

$$f(OPT_{SM}) \geq f(OPT) \geq \tau.$$

Combining the above inequality with the result in (9), we can get that $f(S) \geq \gamma\tau$. Therefore, the algorithm stops before $v_g$ reaches $(1+\alpha)v_{OPT}$. The cost of each partition would satisfy

$$c(S \cap U_j) \leq \beta p_j v_g \leq (1+\alpha)\beta p_j v_{OPT}$$

The proof is completed by setting $v_S = \beta v_g$. $\qquad\square$

## D    Appendix for Section 2.3

In this section, we present the missing proofs of the theoretical results of `Block-Fair-Bi` from Section 2.3. The theorem statement is provided in Theorem 2.8.

**Theorem 2.8.** *Suppose that* `Block-Fair-Bi` *is run for an instance of SMF, then* `Block-Fair-Bi` *outputs a solution $S$ that satisfies a $\left(1-\varepsilon, \frac{\ln\frac{1}{\varepsilon}}{\ln 2}\right)$-bicriteria approximation guarantee in at most $O\left(n\kappa\ln(1/\varepsilon)\right)$ queries of $f$.*

*Proof.* Denote the solution set after the $i$-th chunk as $S_i$, and we denote the subset $B$ after the $i$-th chunk as $B_i$, then it follows that $S_i = S_{i-1} \cup \{B_i\}$. We can prove the following lemma.

**Lemma D.1.** *For any $i \leq \frac{\ln 1/\epsilon}{\ln 2}$, the solution set $S_i$ satisfies that*

$$f(S_i) - f(S_{i-1}) \geq f(OPT) - f(S_i)$$

*and that*

$$|S_i \cap U_c| \leq u_c i,$$
$$\sum_{c \in [N]} \max\{|S_i \cap U_c|, l_c i\} \leq ki.$$

*Proof.* Let us denote the solution set after adding the $j$-th element to the solution set $S$ during the $i$-th chunk as $S_{i,j}$. In addition, we denote the $j$-th element adding to $B_i$ as $b_j$, and that $B_{i,j} = (b_1, \ldots, b_{j-1})$. By the definition of matroid, there exists a mapping from the set $B$ to the optimal solution $OPT = \{o_1, ..., o_\kappa\}$ s.t. $B_{i,j} \cup \{o_j\} \in \mathcal{P}_{fair}$.

$$f(S_{i,j}) - f(S_{i,j-1}) \geq f(S_{i,j-1} \cup \{o_i\}) - f(S_{i,j-1}) \geq \Delta f(S_{i,\kappa}, o_i)$$

Summing over all $j$, it follows that

$$f(S_{i,\kappa}) - f(S_{i,0}) \geq \sum_{i=1}^{\kappa} \Delta f(S_{i,\kappa}, o_i) \geq f(OPT) - f(S_{i,0}).$$

Since $f(S_{i,\kappa}) = f(S_{i+1,0}) = f(S_{i+1})$,

$$f(S_{i+1}) - f(S_i) \geq f(OPT) - f(S_{i+1})$$

Next, we prove the result on the size of the solution set $S$. When $i = 1$, $S_1 = B_1$. By the fact that $B_1 \in \mathcal{M}_{fair}$, we have that the result in the lemma holds. Let us assume that the result in the lemma holds for $i$. Then for $i+1$, we have that $|S_{i+1} \cap U_c| = |S_i \cup B_i \cap U_c| \leq |S_i \cap U_c| + |B_i \cap U_c| \leq u_c(i+1)$. For the total cardinality constraint,

$$\max\{|S_{i+1} \cap U_c|, l_c(i+1)\} \leq \max\{|S_i \cap U_c| + |B_i \cap U_c|, l_c(i+1)\}$$
$$\leq \max\{|S_i \cap U_c|, l_c i\} + \max\{|B_i \cap U_c|, l_c\}$$

where the first inequality comes from the fact that $S_{i+1} = S_i \cup B_i$. The second inequality is due to the inequality of $\max\{a + b, c + d\} \leq \max\{a, c\} + \max\{b, d\}$. It then follows that

$$\sum_{c \in [N]} \max\{|S_{i+1} \cap U_c|, l_c(i+1)\} \leq \sum_{c \in [N]} \max\{|S_i \cap U_c|, l_c i\} + \sum_{c \in [N]} \max\{|B_i \cap U_c|, l_c\} \leq \kappa(i+1)$$

$\square$

Next, by leveraging this Lemma D.1, we can prove the results in Theorem 2.8. Denote $\phi = \frac{\ln 1/\epsilon}{\ln 2}$, then by the Lemma, we have that

$$|S_\phi \cap U_c| \leq u_c \phi,$$
$$\sum_{c \in [N]} \max\{|S_\phi \cap U_c|, l_c \phi\} \leq k\phi.$$

Since $f(S_i) \geq \frac{f(OPT) + f(S_{i-1})}{2}$, by induction, it follows that

$$f(S_\phi) \geq (1 - \frac{1}{2^\phi}) f(OPT) = (1 - \epsilon) f(OPT).$$

$\square$

# E SUBMODULAR MAXIMIZATION UNDER PARTITION MATROID CONSTRAINT

In the previous sections and in the main paper, we demonstrated that block-greedy algorithms can be effective for solving submodular cover problems under partition-based constraints. Interestingly, this block-greedy approach also proves to be valuable in designing algorithms for submodular maximization problems. In this section, we introduce `Block-Greedy`, a novel algorithmic framework tailored for submodular maximization subject to a partition matroid constraint.

`Block-Greedy` proceeds by greedily adding blocks—i.e., sets of elements—to the solution. Our algorithms improve upon existing methods in both solution quality and query complexity.

This section is structured as follows. We first present our main results in Sections E.1, E.2, and E.3. Section E.1 introduces the block-greedy framework that underpins the algorithms discussed throughout. Then, we address two specific settings: monotone submodular maximization with a partition matroid constraint (monotone SMP) in Section E.2, and nonmonotone submodular maximization with a partition matroid constraint (nonmonotone SMP) in Section E.3.

Finally, we include additional content and discussions in Section E.4 and Section E.5. Section E.4 provides the missing discussion and proof of Theorem E.2 from Section E.2.1, while Section E.5 elaborates on omitted content from Section E.2.2.

---

**Algorithm 6** `Block-Greedy`

---

1: **Input:** Partitions of the ground set $U_1, U_2, ..., U_N$, problem definition and parameters
2: **Output:** $S \subseteq U$
3: $S \leftarrow \emptyset$
4: **for** $i = 1$ **to** $\phi$ **do**
5:     **for** $j = 1$ **to** $N$ **do**
6:         `Greedy-Subroutine` $(S, i, j)$
7: **return** $S$

---

### E.1 BLOCK GREEDY FRAMEWORK

The `Block-Greedy` algorithm serves as the core framework for most of our proposed algorithms, except for the `Block-Fair-Bi` algorithm used in the Fair Submodular Cover problem. `Block-Greedy` repeatedly runs a greedy subroutine. In each of the subroutines, a "block" of elements is added into the final solution from each of the partitions of the universe $U$. The value of the parameter $\phi$ and the subroutine `Greedy-Subroutine` are problem-specific and vary depending on the submodular optimization problem being solved. The pseudocode for `Block-Greedy` is in Algorithm 6. In the following part, we introduce the subroutine algorithm for different problems and present the analysis for these proposed algorithms.

### E.2 MONOTONE SMP

We first consider the classic problem of Monotone Submodular Maximization with a Partition Matroid Constraint (SMP). Given positive integers $k_1, \cdots k_N$ such that $k_j \leq |U_j|$ for any $j \in [N]$, the partition matroid constraint is defined as $\mathcal{P} = \{S \subseteq U : |S \cap U_j| \leq k_j, \forall j \in [N]\}$. The monotone SMP is defined to find the set $\arg\max_{S \in \mathcal{P}} f(S)$ for a monotone, submodular objective function $f$. Before presenting our algorithm, we illustrate the intuitions and benefits of our proposed algorithm in contrast to the standard greedy algorithm through a tight hardness result.

#### E.2.1 TIGHT EXAMPLES

The standard greedy algorithm iteratively selects the element with the highest marginal gain while maintaining feasibility. It is well-known that this algorithm achieves a $1/2$-approximation ratio for monotone submodular maximization with general matroid constraint. Despite partition matroids being a simpler special case, in the theorem below, we prove this ratio is tight by constructing a class of instances where the standard greedy algorithm cannot achieve an approximation ratio better than $1/2$.

**Theorem E.1.** *For any given positive integers $k_1, ..., k_N$, there exists an instance of monotone SMP with size constraints $k_1, ..., k_N$, i.e.,*

$$\max_{S \in \mathcal{P}} f(S)$$

*where $\mathcal{P} := \{S \subseteq U : |S \cap U_i| \leq k_i\}$, such that the best approximation ratio achievable by the standard greedy algorithm is $1/2$.*

We defer the detailed proof of Theorem E.1 to the supplementary material in Section E.4. We give a brief illustration of the proof by constructing a toy example. Suppose $U = [8]$ which is split into two groups $U_1 = \{1, 2, 3, 4\}$, $U_2 = \{5, 6, 7, 8\}$. Let $t : U \rightarrow M$ be a function that assigns tags to each element in the universe: $t(1) = t(5) = t(7) = t(8) = "a"$, $t(2) = t(6) = "b"$, $t(3) = "c"$, and $t(4) = "d"$. Define a set cover function $f$ that maps a subset to the number of unique tags covered, $f(S) = |\cup_{s \in S} t(s)|$. Here, the partition matroid is defined by $k_1 = k_2 = 2$. In this case, the standard greedy algorithm might first select elements 1 and 2. Subsequently, all remaining elements either become infeasible or have zero marginal gain, yielding a solution set $S$ with $f(S) = 2$, whereas the optimal solution set $OPT = 3, 4, 5, 6$ achieves $f(OPT) = 4$. Therefore, $f(S) = f(OPT)/2$. Thus we can conclude the proof.

This example highlights a key limitation of the standard greedy algorithm: it greedily adds the element with the highest marginal gain by searching over all feasible elements in each step, which can lead certain partitions to quickly reach their cardinality limits, making remaining elements in

those subgroups infeasible for later selections. This strategy prevents the standard greedy algorithm from achieving a better approximation ratio.

In fact, in most of the continuous methods developed in existing works (Badanidiyuru and Vondrák, 2014; Calinescu et al., 2011), the key idea for achieving the optimal approximation ratio of $1 - 1/e$ is by incrementally increasing some coordinates by small fractions in each step. This technique ensures that all of the elements in the ground set $U$ remain feasible throughout most of the algorithm's execution. Inspired by this insight, the `Block-Greedy` algorithm carefully balances the number of elements being added to the solution set across different partitions during each round. In particular, the number of elements selected by `Block-Greedy` at each round within each partition $i$ is proportional to the budget capacity $k_i$ of the partition to ensure the feasibility of the elements for the majority of the algorithm's runtime, thereby effectively addressing this limitation. We provide a detailed description of our proposed algorithm for monotone SMP in the next section.

### E.2.2 Subroutine Algorithm for Monotone SMP

We propose the subroutine algorithm `Greedy-Subroutine-Mono` (Algorithm 7), to be used in `Block-Greedy` along with the parameter $\phi = \lfloor \sqrt{\min_{i \in [N]} k_i} \rfloor - 1$, and show that it can be used to achieve an approximate solution that is at least as good as the standard greedy and often strictly better. Further, `Block-Greedy` makes fewer queries to $f$, depending on the structure of the partition matroid constraint. Here the parameter $r_j$ is defined as $r_j := \lfloor k_j/\phi \rfloor$ for each $j \in [N]$.

---

**Algorithm 7** `Greedy-Subroutine-Mono` $(S, i, j)$

---

1: **Input:** $S, i, j$
2: **for** $l = 1$ **to** $r_j$ **do**
3: $\quad S \leftarrow S \cup \arg\max_{x \in U_j} \Delta f(S, x)$

---

**Theorem E.2.** *Suppose that* `Block-Greedy` *is run for an instance of monotone SMP with the subroutine algorithm* `Greedy-Subroutine-Mono` *as described in Algorithm 7, then* `Block-Greedy` *outputs a solution set $S$ that satisfies an approximation ratio of $1 - 1/e - \frac{1}{\phi+1}$ where $\phi = \lfloor \sqrt{\min_{i \in [N]} k_i} \rfloor - 1$.*

Intuitively, `Block-Greedy` achieves an approximation close to $1 - 1/e$ when the parameters $k_i$ are large. The reason that the term involving $\phi$ arises in Theorem E.2 is because there are a total of $\phi$ rounds in the outer loop of `Block-Greedy`. In particular, if $\phi \geq 7$, then the approximation guarantee described in Theorem E.2 is strictly better than 1/2. To further ensure the bound is better than 1/2, we can greedily add new elements with maximum marginal gain to the returned solution by `Block-Greedy` algorithm until the cardinality of the solution set reaches the rank of the partition matroid, in which case the approximation ratio of `Block-Greedy` is $\max\{1/2, 1 - 1/e - \frac{1}{\phi+1}\}$ (see Appendix E.5 for proof).

Notably, the difference between the approximation ratio for `Block-Greedy` and the optimal result of $1 - 1/e$ is bounded by $\mathcal{O}(\frac{1}{\sqrt{k_{\min}}})$ where $k_{\min} = \min_{i \in [N]} k_i$. In particular, in some cases where $k_1 = k_2 = \cdots = k_N$, the bound can be improved further to $\mathcal{O}(\frac{1}{k_1})$, as shown in the following corollary:

**Corollary E.3.** *Suppose* `Block-Greedy` *with the* `Greedy-Subroutine-Mono` *subroutine is run for instance of monotone SMP with $k_1 = k_2 = ... = k_N$. If we set $\phi = k_1$ and $r_j = 1$ for each $j$, then* `Block-Greedy` *outputs a solution set $S$ with an approximation ratio of $1 - 1/e - 1/k_1$.*

The corollary can be proved by applying Theorem E.8, which is presented and analyzed in Appendix E.5.

An additional important benefit to `Block-Greedy` compared to the standard greedy algorithm is that its query complexity is potentially much better. This improvement arises because `Block-Greedy` selects elements with maximum marginal gain within one partition rather than over the entire universe $U$. In particular, the query complexity of `Block-Greedy` is upper bounded by $\sum_{i \in [N]} |U_i| k_i \leq n(\sum_{i \in [N]} k_i)$.

Next, we present the proof of the theorem. First of all, we prove the result in Lemma E.4.

**Lemma E.4.** *Let us define the partition matroid of $\{S \subseteq U : |S \cap U_j| \leq r_j\phi\}$ as $\mathcal{P}'$, and we define the optimal solution of the problem $\max_{S \in \mathcal{P}'} f(S)$ as $OPT'$. Let us denote the input and output of Algorithm 7 as $S$ and $S'$ respectively, then it follows that*

$$f(S') - f(S) \geq \frac{\Delta f(S', OPT_j)}{\phi},$$

*where $OPT_j = OPT' \cap U_j$.*

*Proof.* For notation simplicity, we define the solution set after the $l$-th step in the for loop of Algorithm 7 as $S^{(l)}$. By the greedy selection step in Line 3, we have that for any $o \in OPT_j := OPT' \cap U_j$,

$$f(S^{(l)}) - f(S^{(l-1)}) \geq \Delta f(S^{(l-1)}, o).$$

Therefore,

$$
\begin{aligned}
f(S^{(l)}) - f(S^{(l-1)}) &\geq \frac{\sum_{o \in OPT_j} \Delta f(S^{(l-1)}, o)}{|OPT_j|} \\
&\geq \frac{\sum_{o \in OPT_j} \Delta f(S^{(l-1)}, o)}{r_j \phi} \\
&\geq \frac{\sum_{o \in OPT_j} \Delta f(S^{(r_j)}, o)}{r_j \phi} \\
&\geq \frac{\Delta f(S^{(r_j)}, OPT_j)}{r_j \phi},
\end{aligned}
$$

where the second inequality follows from the fact that $OPT' \in \mathcal{P}'$, and therefore $|OPT_j| \leq r_j\phi$. Summing over all $l \in [r_j]$, it follows that

$$f(S^{(r_j)}) - f(S^{(0)}) \geq \frac{\Delta f(S^{(r_j)}, OPT_j)}{\phi}.$$

Notice that $S^{(0)}$ is the input of the algorithm and $S^{(l)}$ is the output of the algorithm, so we can prove the result. $\qquad\square$

With the result in Lemma E.4, we can prove the result in Theorem E.2.

*Proof.* Let $S_{i,j}$ represent the solution set after executing the subroutine algorithm `Greedy-Subroutine-Mono` on the $j$-th subgroup during the $i$-th iteration of the outer for loop in Line 4 in Algorithm 6, and we define $S_i$ as the solution set after completing the $i$-th round of the outer for loop in Algorithm 6, i.e., $S_i = S_{i,N}$. Then by the result in Lemma E.4, we have that we have that

$$f(S_{i,j}) - f(S_{i,j-1}) \geq \frac{\Delta f(S_{i,j}, OPT_j)}{\phi}$$

Since $S_{i,j} \subseteq S_{i,N}$ for any $j \in [N]$, by submodularity, we have that $\Delta f(S_i, OPT_j) = \Delta f(S_{i,N}, OPT_j) \leq \Delta f(S_{i,j}, OPT_j)$. Then

$$f(S_{i,j}) - f(S_{i,j-1}) \geq \frac{\Delta f(S_i, OPT_j)}{\phi}$$

Summing over all $j$, it then follows that

$$
\begin{aligned}
f(S_{i,N}) - f(S_{i,0}) &\geq \frac{\sum_{j \in [N]} \Delta f(S_i, OPT_j)}{\phi} \\
&\geq \frac{\Delta f(S_i, OPT')}{\phi},
\end{aligned}
$$

---

**Algorithm 8** `Greedy-Subroutine-Nonmono` $(S, i, j)$

---

1: **Input:** $S, i, j$
2: **for** $l = 1$ **to** $r_j$ **do**
3:      Let $M \subseteq U/S$ be a set of size $r_j\phi$ maximizing $\sum_{x \in M} \Delta f(S, x)$.
4:      $u \leftarrow$ uniformly sample an element from $M$
5:      $S \leftarrow S \cup \arg\max_{x \in U_j} \Delta f(S, x)$

---

where the last inequality follows from submodularity and the fact that $OPT' = \cup_{j \in [N]} OPT_j$. Notice that here $S_{i,N}$ is equivalent to $S_i$, and that $S_{i,0}$ is equivalent to $S_{i-1}$. Then we get

$$f(S_i) - f(S_{i-1}) \geq \frac{f(OPT' \cup S_i) - f(S_i)}{\phi}$$
$$\geq \frac{f(OPT') - f(S_i)}{\phi}.$$

By rearranging the inequality and by induction, we have that

$$f(S_\phi) - f(\emptyset) \geq (1 - (\frac{\phi}{\phi + 1})^\phi) f(OPT').$$

By the definition of $\phi$ that $\phi = \lfloor \sqrt{\min_{i \in [N]} k_i} \rfloor - 1$, we have that $k_i - \phi \lfloor k_i/\phi \rfloor \leq \lfloor k_i/\phi \rfloor$ for any $i \in [N]$. By Lemma E.7, it follows that

$$\max_{S \in \mathcal{P}'} f(S) \geq \frac{\phi}{\phi + 1} \max_{S \in \mathcal{P}} f(S).$$

Therefore,

$$f(S_\phi) \geq (1 - (\frac{\phi}{\phi + 1})^\phi) f(OPT')$$
$$\geq (1 - (\frac{\phi}{\phi + 1})^\phi)(\frac{\phi}{\phi + 1}) f(OPT)$$
$$\geq (1 - e^{-1} - \frac{1}{\phi + 1}) f(OPT).$$

$\square$

### E.3 NONMONOTONE SMP

In this section, we propose the algorithm for the problem of nonmonotone Submodular Maximization over Partition matroid (SMP). The proposed algorithm follows the framework in Algorithm 6 with $\phi = \lfloor \sqrt{\min_{i \in [N]} k_i} \rfloor - 1$, and the subroutine algorithm `Greedy-Subroutine-Nonmono` is described in Algorithm 8. Here the parameter $r_j := \lfloor k_j/\phi \rfloor$. The algorithm uniformly selects an element from the set of elements with the top $r_j\phi$ marginal gain to add to the solution set. The intuition behind the `Greedy-Subroutine-Nonmono` algorithm is similar to that of the Random Greedy algorithm proposed in Buchbinder et al. (2014). However, in `Greedy-Subroutine-Nonmono`, the size of the candidate set considered for inclusion in the solution is adjusted to $r_j\phi$ to ensure an important result that $\mathbb{E}[f(S_i \cup OPT')] \geq (1 - \frac{1}{\phi})^i f(OPT')$ where $\mathcal{P}' = \{S \subseteq U : |S \cap U_i| \leq r_i\phi, \forall i \in [N]\}$ and $OPT' = \arg\max_{S \in \mathcal{P}'} f(S)$.

Below we present the main result of `Block-Greedy` for the problem of nonmonotone SMP.

**Theorem E.5.** *Suppose that `Block-Greedy` is run for an instance of nonmonotone SMP with the subroutine algorithm `Greedy-Subroutine-Nonmono` as described in Algorithm 7, then `Block-Greedy` outputs a solution $S$ that satisfies an approximation ratio of $\frac{1}{e} - \frac{3}{e(\phi+1)}$ where $\phi = \sqrt{\min_{i \in [N]} k_i} - 1$ in expectation.*

Notice that the approximation ratio is close to $1/e$, which matches the bound of the random greedy algorithm for submodular maximization under cardinality constraint, with the difference bounded by

$\mathcal{O}(\frac{1}{\sqrt{k_{\min}}})$, where $k_{\min} = \min_{i \in [N]} k_i$. In this sense, the proposed algorithm achieves an approximation ratio for Nonmonotone SMP that bridges the gap between submodular maximization over cardinality constraint and partition matroid constraint. The proof of Theorem E.5 is provided below.

Let us define $\mathcal{P}' = \{S \subseteq U : |S \cap U_i| \leq r_i \phi, \forall i \in [N]\}$ and we denote the optimal solution set of $OPT' = \arg \max_{S \in \mathcal{P}'} f(S)$. First of all, we prove the following lemma for the subroutine algorithm `Greedy-Subroutine-Nonmono`.

**Lemma E.6.** *Let us denote the input and output of Algorithm 8 as $S$ and $S'$ respectively, then it follows that*

$$\mathbb{E}[f(S') - f(S)] \geq \frac{\mathbb{E}[\Delta f(S', OPT_j)]}{\phi},$$

*where $OPT_j = OPT' \cap U_j$.*

*Proof.* Let us denote the solution set after adding the $l$-th element as $S^{(l)}$. Similar to the proof of Theorem E.2 for the monotone SMP, we have that

$$\mathbb{E}[f(S^{(l)}) - f(S^{(l-1)})] \geq \frac{\sum_{a \in OPT' \cap U_j} \Delta f(S^{(l-1)}, a)}{r_j \phi}.$$

By submodularity, we would have that

$$\mathbb{E}[f(S^{(l)}) - f(S^{(l-1)})] \geq \mathbb{E}\left[\frac{\sum_{a \in OPT' \cap U_j} \Delta f(S^{(l-1)}, a)}{r_j \phi}\right]$$

$$\geq \mathbb{E}\left[\frac{\sum_{a \in OPT' \cap U_j} \Delta f(S', a)}{r_j \phi}\right].$$

By summing over all $l \in [r_j]$, it follows that

$$\mathbb{E}[f(S') - f(S)] \geq \mathbb{E}\left[\frac{\sum_{a \in OPT' \cap U_j} \Delta f(S', a)}{\phi}\right]$$

$$\geq \mathbb{E}\left[\frac{\Delta f(S', OPT_j)}{\phi}\right].$$

$\square$

Next, leveraging the result in Lemma E.6, we prove the approximation ratio in Theorem E.5.

*Proof.* Similar to the proof of Theorem E.2, we denote the solution set after running the subroutine algorithm in $j$-th subgroup during the $i$-th round of the outer for loop in Line 4 in Algorithm 6 as $S_{i,j}$, and we define the solution set after completing the $i$-th round in Algorithm 6 as $S_i$. From Lemma E.6, we have that

$$\mathbb{E}[f(S_{i,j}) - f(S_{i,j-1})] \geq \frac{\mathbb{E}[\Delta f(S_{i,j}, OPT_j)]}{\phi}$$

$$\geq \frac{\mathbb{E}[\Delta f(S_i, OPT_j)]}{\phi},$$

By summing over all $j \in [N]$, we have

$$\mathbb{E}[f(S_i) - f(S_{i-1})] \geq \frac{\sum_{j=1}^{N} \mathbb{E}[\Delta f(S_i, OPT_j)]}{\phi}$$

$$\geq \frac{\mathbb{E}[\Delta f(S_i, OPT')]}{\phi}.$$

Then it follows that

$$\mathbb{E}[f(S_i) - f(S_{i-1})] \geq \mathbb{E}\left[\frac{f(S_i \cup OPT') - f(S_i)}{\phi}\right].$$

Notice that by the greedy selection step, for each group $j$ and each element $a \in OPT' \cap U_j$ appears in $S_i$ with probability at most $1 - (1 - \frac{1}{r_j \phi})^{r_j i}$. Since $(1 - \frac{1}{x})^x$ increases with $x$ in the range of $[1, +\infty)$, we have that $(1 - \frac{1}{r_j \phi})^{r_j \phi} \geq (1 - \frac{1}{\phi})^\phi$. Therefore, we would get $1 - (1 - \frac{1}{r_j \phi})^{r_j i} \leq 1 - (1 - \frac{1}{\phi})^i$. From Lemma 2.2 in Buchbinder et al. (2014), we can conclude that

$$\mathbb{E}[f(S_i \cup OPT')] \geq (1 - \frac{1}{\phi})^i f(OPT').$$

By rearranging the above inequality, we can get that

$$\mathbb{E}[f(S_i)] \geq \frac{\phi}{\phi + 1} \mathbb{E}[f(S_{i-1})] + \frac{1}{\phi + 1}(1 - \frac{1}{\phi})^i f(OPT').$$

By induction, we have that the output solution set satisfies that

$$\mathbb{E}[f(S)] = \mathbb{E}[f(S_\phi)]$$

$$\geq \frac{1}{\phi + 1} \sum_{i=1}^{\phi} (\frac{\phi}{\phi + 1})^{\phi - i}(1 - \frac{1}{\phi})^i f(OPT')$$

$$\geq \frac{1}{\phi + 1} \sum_{i=1}^{\phi} (\frac{\phi - 1}{\phi})^{\phi - i}(1 - \frac{1}{\phi})^i f(OPT')$$

$$\geq \frac{\phi}{\phi + 1}(1 - \frac{1}{\phi})^\phi f(OPT') \geq \frac{1}{e}(1 - \frac{2}{\phi + 1})f(OPT').$$

where the last inequality follows from the fact that $(1 - \frac{1}{\phi})^{\phi - 1} \geq e^{-1}$ for any $\phi > 1$. From the definition that $\phi = \lfloor \sqrt{\min_{i \in [N]} k_i} \rfloor - 1$, it then follows that $k_i - \phi \lfloor k_i / \phi \rfloor \leq \lfloor k_i / \phi \rfloor$ for any $i \in [N]$. Therefore, from the result in Lemma E.7, we get that

$$f(OPT') \geq \frac{\phi}{\phi + 1} f(OPT) \tag{10}$$

where $OPT$ is the optimal solution to the submodular maximization problem $\arg\max_{S \in \mathcal{P}} f(S)$. By combining (3) and (10) together, we can prove the result in the Lemma. $\qquad\square$

**Lemma E.7.** *Suppose the ground set $U$ is divided into $N$ disjoint subgroups $U_1$, $U_2$,..., $U_N$, then for any partition matroid $\mathcal{P} = \{S \subseteq U : |S \cap U_j| \leq k_j, \forall j \in [N]\}$, let us define the matroid $\mathcal{P}' = \{S \subseteq U : |S \cap U_j| \leq \lfloor \frac{k_j}{c} \rfloor c, \forall j \in [N]\}$ for some positive integer $c$. If for any $j \in [N]$, it satisfies that $\lfloor \frac{k_j}{c} \rfloor \geq k_j - c\lfloor \frac{k_j}{c} \rfloor$, then it follows that for any submodular function $f$, we have*

$$\max_{S \in \mathcal{P}'} f(S) \geq \frac{c}{c + 1} \max_{S \in \mathcal{P}} f(S).$$

*Proof.* For notation simplicity, we also define $r_j = \lfloor \frac{k_j}{c} \rfloor$ where $j \in [N]$, and we define two matroids $\mathcal{P}_0 = \{S \subseteq U : |S \cap U_j| \leq r_j, \forall j \in [N]\}$ and $\mathcal{P}_1 = \{S \subseteq U : |S \cap U_j| \leq r_j(c + 1), \forall j \in [N]\}$. Let us denote the optimal solution of $\max_{S \in \mathcal{P}_1} f(S)$ as $OPT''$. Then by definition, we can see that $OPT''$ can be divided into $(c + 1)$ disjoint subsets $O_1, ..., O_{c+1}$ such that each $O_i \in \mathcal{P}_0$. Without loss of generality, we assume that the subsets are chosen greedily such that the index satisfies $\Delta f(\cup_{j \in [i-1]} O_j, O_i) \geq \Delta f(\cup_{j \in [i-1]} O_j, O_l)$ for any $1 \leq i \leq c$ and $l > i$. It then follows that by submodularity, $\Delta f(\cup_{j \in [i-1]} O_j, O_i) \geq \Delta f(\cup_{j \in [i-1]} O_j, O_l) \geq \Delta f(\cup_{j \in [l-1]} O_j, O_l)$ for any $l > i$. Therefore,

$$f(OPT'') - f(\cup_{j \in [c]} O_j) = \Delta f(\cup_{j \in [c]} O_j, O_{c+1})$$

$$\leq \frac{\sum_{i \in [c]} \Delta f(\cup_{j \in [i-1]} O_j, O_i)}{c}$$

$$= \frac{f(\cup_{j \in [c]} O_j) - f(\emptyset)}{c}.$$

By rearranging the above inequality, we would get that $f(\cup_{j \in [c]} O_j) \geq \frac{c}{c+1} f(OPT'')$. Since $\cup_{j \in [c]} O_j \in \mathcal{P}'$, then we have that $\max_{S \in \mathcal{P}'} f(S) \geq \frac{c}{c+1} f(OPT'')$. Notice that $\lfloor \frac{k_j}{c} \rfloor (c + 1) \geq k_j$ implies that for any subset $S \in \mathcal{P}$, it also satisfies that $S \in \mathcal{P}_1$. Therefore, $\max_{S \in \mathcal{P}_1} f(S) \geq \max_{S \in \mathcal{P}} f(S)$ and we can conclude the proof.

$\qquad\square$

### E.4 Proof of Theorem E.1

In this section, we present the omitted proof of Theorem E.1. To prove the theorem, we construct a class of instances to demonstrate that the standard greedy algorithm can't achieve an approximation ratio better than $1/2$. We begin by presenting relevant definitions of the set functions and constraints, followed by a formal description of the hardness instance.

Suppose the ground set $U$ is partitioned into $N$ disjoint subsets $U_1, U_2, \ldots, U_N$, with each subset $U_i$ containing $2k_i$ elements. Define a set function $t : U \to 2^M$, where $M$ is a finite set, and let $c : 2^M \to \mathbb{R}_+$ be a non-negative, monotone, modular function. We define the submodular function $f : 2^U \to \mathbb{R}_+$ as follows:

$$f(S) = c\left(\bigcup_{s \in S} t(s)\right) = \sum_{x \in \bigcup_{s \in S} t(s)} c(x).$$

The partition matroid is denoted as $\mathcal{P} = \{S \subseteq U : |S \cap U_i| \leq k_i, \forall i \in [N]\}$. Without loss of generality, we assume the partition matroid constraint parameters satisfy that $k_1 \leq k_2 \leq, \ldots \leq k_N$. For notation simplicity, let us denote the $j$-th element in group $U_i$ as $s_j^{(i)}$. The hardness example is defined as follows. Suppose $\epsilon$ is a constant such that $\epsilon \in (0, 1/2)$, then

1. **Set Assignments in $U_1$:** In the first partition $U_1$, assume that the sets $t(s_{j_1}^{(1)})$ and $t(s_{j_2}^{(1)})$ are disjoint for any distinct $j_1, j_2$, i.e., $t(s_{j_1}^{(1)}) \cap t(s_{j_2}^{(1)}) = \emptyset$.

2. **Function Values in $U_1$:** Set the modular function values for the elements in $U_1$ as follows:
   - For $j \leq k_1$, $c(t(s_j^{(1)})) = \frac{1}{2} + \epsilon$.
   - For $k_1 < j \leq 2k_1$, $c(t(s_j^{(1)})) = \frac{1}{2}$.

3. **Set Assignments and Function Values for $i > 1$:** For each $i_1, i_2$ such that $1 < i_1, i_2 \leq N$, define:
   - If $j \leq k_1$, then $t(s_j^{(i_1)}) = t(s_j^{(i_2)}) \subseteq t(s_j^{(1)})$.
   - If $j > k_1$, then $t(s_j^{(i_1)}) = t(s_j^{(i_2)}) = t(s_1^{(i_1)})$.
   - Set $c(t(s_j^{(i)})) = \frac{1}{2}$ for any $i > 1$ and $j \in [2k_i]$.

Given this construction, the standard greedy algorithm proceeds as follows: for the first $k_1$ steps, the algorithm would add the elements $s_1^{(1)}, s_2^{(1)}, \ldots, s_{k_1}^{(1)}$ from $U_1$, yielding a marginal gain of $\frac{1}{2} + \epsilon$ per step. Thus, the total value after these steps is $\frac{k_1}{2} + k_1 \epsilon$. After the first $k_1$ steps, the algorithm can only select elements from partitions $U_i$ where $i > 1$, with a marginal gain of $0$ at each step due to the structure of $f$ under the current set assignments. Therefore, the value of the submodular objective returned by the standard greedy algorithm is $\frac{k_1}{2} + k_1 \epsilon$.

Next, we consider the optimal solution of the constructed example. Notice that for any of the partitions $U_i$ such that $i > 1$, $f(U_i) = k_1/2$. Besides, by the construction, we have that $\bigcup_{s \in U_1} t(s) = \bigcup_{s \in U_2} t(s)$ for any $i_1, i_2 > 1$. Therefore, $f(\bigcup_{i>1} U_i) = k_1/2$. It then follows that for any subset $S \subseteq U$,

$$f(S \cap \bigcup_{i>1} U_i) \leq k_1/2. \tag{11}$$

Next, we claim that the optimal solution should satisfy that $f(OPT) \leq k_1$. We prove the claim by considering the following cases. For any $S \in \mathcal{P}$, then

1. If $f(S \cap \bigcup_{i>1} U_i) = k_1/2$, which means $\bigcup_{s \in S \cap \bigcup_{i>1} U_i} t(s) = \bigcup_{j \leq k_1} t(s_j^{(2)})$, then for each $j \leq k_1$, the marginal gain of adding the $j$-th element in the first partition to the the set $S \cap \bigcup_{i>1} U_i$ satisfies that $\Delta f(S \cap \bigcup_{i>1} U_i, s_j^{(1)}) = c(t(s_j^{(1)})) - c(t(s_j^{(2)})) = \epsilon$, and for each $j > k_1$, $\Delta f(S \cap \bigcup_{i>1} U_i, s_j^{(1)}) = f(s_j^{(1)}) = 1/2$. It then follows that $\Delta f(S \cap \bigcup_{i>1} U_i, S \cap U_1) \leq \sum_{s \in S \cap U_1} \Delta f(S \cap \bigcup_{i>1} U_i, s) \leq k_1/2$. Therefore, $f(S) = \Delta f(S \cap \bigcup_{i>1} U_i, S \cap U_1) + f(S \cap \bigcup_{i>1} U_i) \leq k_1$.

2. If $f(S \cap \bigcup_{i>1} U_i) < k_1/2$, then we have that there exists at least one element $s_j^{(2)}$ for $j \leq k_1$ such that the set $t(s_j^{(2)}) \notin \bigcup_{s \in S \cap \bigcup_{i>1} U_i} t(s)$ . Let us denote $E = \{j \leq k_1 : t(s_j^{(2)}) \notin S \cap \bigcup_{i>1} U_i\}$. It then follows that $f(S \cap \bigcup_{i>1} U_i) = \frac{k_1 - |E|}{2}$. For each $j \leq k_1$, if $j \in E$, $\Delta f(S \cap \bigcup_{i>1} U_i, s_j^{(1)}) = c(t(s_j^{(1)})) = 1/2 + \epsilon$, if $j \notin E$, then $\Delta f(S \cap \bigcup_{i>1} U_i, s_j^{(1)}) = c(t(s_j^{(1)})) - c(t(s_j^{(2)})) = \epsilon$. For each $j > k_1$, $\Delta f(S \cap \bigcup_{i>1} U_i, s_j^{(1)}) = f(s_j^{(1)}) = 1/2$. Similarly, we have that $\Delta f(S \cap \bigcup_{i>1} U_i, S \cap U_1) \leq \sum_{s_j^{(1)} \in S \cap U_1} \Delta f(S \cap \bigcup_{i>1} U_i, s_j^{(1)}) \leq |E|(1/2 + \epsilon) + (k_1 - |E|)/2 = |E|\epsilon + k_1/2$. Therefore, we can conclude that $f(S) = \Delta f(S \cap \bigcup_{i>1} U_i, S \cap U_1) + f(S \cap \bigcup_{i>1} U_i) \leq k_1 - |E|(1/2 - \epsilon) \leq k_1$.

It then follows that the $f(\text{OPT}) \leq k_1$. Notice that the set $O = \{s_{k_1+1}^{(1)}, ...s_{2k_1}^{(1)}, s_1^{(2)}, ..., s_{k_1}^{(2)}\}$ achieves an objective value of: $f(O) = k_1$. Therefore, $f(\text{OPT}) = k_1$. Consequently, the approximation ratio of the standard greedy algorithm should be $\frac{k_1/2 + k_1\epsilon}{k_1} = 1/2 + \epsilon$. When $\epsilon$ approaches 0, then the approximation ratio goes to $1/2$.

### E.5   DISCUSSION ON THEOREM E.2

In this portion of the appendix, we illustrate the results of Theorem E.2. First of all, we discuss the difference between the approximation ratio of our proposed algorithm Block-Greedy and the optimal approximation ratio $1 - 1/e$ achieved by the previous continuous method. In particular, the difference is $\mathcal{O}(\frac{1}{\sqrt{k_{\min}}})$ with $k_{\min} = \min_{i \in [N]} k_i$. In fact, we notice that this difference results from the fact that it scales in the order of $\mathcal{O}(\frac{1}{\phi})$. In the algorithm Block-Greedy with Greedy-Subroutine-Mono as the subroutine in Section E.2, $\phi$ is set to be $\phi = \lfloor \sqrt{k_{\min}} \rfloor - 1$, which is designed to bound the difference between $k_i$ and $\lfloor \frac{k_i}{\phi} \rfloor \phi$ to ensure that the optimal value of the monotone $\max_{S \in \mathcal{P}'} f(S)$ approximates the optimal value of $\max_{S \in \mathcal{P}} f(S)$ where $\mathcal{P}' := \{S \subseteq U : |S \cap U_i| \leq \lfloor \frac{k_i}{\phi} \rfloor \phi, \forall i \in [N]\}$ and $\mathcal{P} := \{S \subseteq U : |S \cap U_i| \leq k_i, \forall i \in [N]\}$.

This motivates the following result: in some cases, if we can design the parameter $\phi$ such that $\lfloor \frac{k_i}{\phi} \rfloor = \frac{k_i}{\phi}$ for any $i \in [N]$, then the partition matroid $\mathcal{P} = \mathcal{P}'$ and we don't need to bound the difference of $\max_{S \in \mathcal{P}'} f(S)$ and $\max_{S \in \mathcal{P}} f(S)$. Therefore, we can further refine the difference between the approximation ratio of Block-Greedy and the optimal result of $1 - 1/e$. The result is stated as follows.

**Theorem E.8.** *Suppose that $gcd(k_1, k_2, \ldots, k_N) = c$, and that Block-Greedy with Greedy-Subroutine-Mono as a subroutine and $\phi = c$ and $r_j = k_j/c$ for each $j \in [N]$ is run for an instance of monotone SMP over partition matroid $\mathcal{P} := \{S \subseteq U : |S \cap U_i| \leq k_i, \forall i \in [N]\}$, then Block-Greedy outputs a solution set $S$ that satisfies an approximation ratio of $1 - 1/e - 1/c$.*

*Proof.* Following the similar proof of Theorem E.2, we can get that the output solution set $S$ satisfies

$$f(S) - f(\emptyset) \geq (1 - (\frac{\phi}{\phi + 1})^\phi) f(OPT'),$$

where $OPT' = \arg \max_{S \in \mathcal{P}'} f(S)$ and $\mathcal{P}' := \{S \subseteq U : |S \cap U_i| \leq r_i \phi, \forall i \in [N]\}$. By the assignment of $r_i$ and $\phi$ in this case, we can get that $r_i \phi = k_i$. It then follows that $\mathcal{P}' = \mathcal{P}$ and that $OPT'$ is also the optimal solution to our problem. Therefore,

$$f(S) \geq (1 - (\frac{\phi}{\phi + 1})^\phi) f(OPT)$$
$$\geq (1 - 1/e - 1/\phi) f(OPT) = (1 - 1/e - 1/c) f(OPT).$$

$\square$

In particular, if $c = \mathcal{O}(k_{\min})$ such as in the case where $k_1 = k_2 =, ... = k_N = k$, we have that the approximation ratio is $1 - 1/e - 1/k$. Therefore, the difference between the approximation ratio and the optimal one is decreased to $\mathcal{O}(\frac{1}{k_{min}})$. The result is stated in Corollary E.3

Next, we prove that we can improve the approximation ratio of `Block-Greedy` algorithm by adding more elements to the solution set. First, we notice that there are two drawbacks of the proposed algorithm `Block-Greedy` compared with the standard greedy algorithm. First of all, the approximation ratio of $1 - 1/e - \frac{1}{\lfloor \sqrt{\min_{i \in [N]} k_i} \rfloor}$ is only better than the approximation ratio of the standard greedy algorithm, which is $1/2$, when the capacity $k_i$ within partition $U_i$ satisfies that $k_i \geq 64$ for each $i \in [N]$.

Second, the output solution satisfies that $|S \cap U_i| \leq r_i \phi$ for each $i \in [N]$. Notice that $r_i \phi \leq k_i$. If $r_i \phi < k_i$, we can add more elements to the solution set $S$ until it reaches the full rank of the partition matroid. Since the objective function is monotone, we can see that adding more elements would not incur a decrease in the marginal gain. In the following part, we claim that if the standard greedy procedure (Algorithm 9) is applied to the output of `Block-Greedy` with `Greedy-Subroutine-Mono` as the subroutine, the resulting solution set achieves an approximation ratio of $\max\{1/2, 1 - 1/e - \frac{1}{\phi+1}\}$.

---

**Algorithm 9** `Greedy`

---

1: **Input:** the output solution set $S$ obtained by running `Block-Greedy` with `Greedy-Subroutine-Mono` as the subroutine and $\phi = \lfloor \sqrt{\min_{i \in [N]} k_i} \rfloor - 1$ and $r_j := \lfloor k_j/\phi \rfloor$
2: **Output:** $A \in U$
3: $A \leftarrow S$
4: **while** $\exists x$ such that $A \cup \{x\} \in \mathcal{P}$ **do**
5: $\quad A \leftarrow A \cup \arg\max_{x \in U, A \cup \{x\} \in \mathcal{P}} \Delta f(A, x)$
$\quad$ **return** $A$

---

**Theorem E.9.** *Suppose we run the standard greedy algorithm in Algorithm 9 with input being the output solution set of the `Block-Greedy` algorithm, then the output solution set achieves an approximation ratio of $\max\{1/2, 1 - 1/e - \frac{1}{\phi+1}\}$ where $\phi = \lfloor \sqrt{\min_{i \in [N]} k_i} \rfloor - 1$.*

*Proof.* First of all, notice that $S \subseteq A$, by the result of Theorem E.2, we can see that $f(S) \geq (1 - 1/e - \frac{1}{\phi+1})f(OPT)$. Since $f$ is monotone, $f(A) \geq f(S) \geq (1 - 1/e - \frac{1}{\phi+1})f(OPT)$. Then to prove the result in the Theorem E.9, it suffices to prove that $f(A) \geq f(OPT)/2$. Here we use the same notations as in the proof of Theorem E.2, which means that we define the partition matroid of $\{S \subseteq U : |S \cap U_j| \leq r_j \phi\}$ as $\mathcal{P}'$, and we define the optimal solution of the problem $\max_{S \in \mathcal{P}'} f(S)$ as $OPT'$. Denote the solution set after completing the $i$-th round of the outer for loop in Line 4 in Algorithm 6 as $S_i$. Following the similar idea in the proof of Theorem E.2, we can see that for any $O \in \mathcal{P}'$, it holds that

$$f(S_i) - f(S_{i-1}) \geq \frac{\Delta f(S_i, O)}{\phi}$$
$$\geq \frac{\Delta f(S, O)}{\phi},$$

where the last inequality follows from submodularity and the fact that $S_i \subseteq S_\phi = S$. Summing over all $i$, then we get

$$f(S) - f(\emptyset) \geq \Delta f(S, O). \tag{12}$$

Let us define the partition matroid $\mathcal{P}'' := \{S \subseteq U : |S \cap U_i| \leq k_i - r_i \phi, \forall i \in [N]\}$. Let us define the solution set $A$ before the $i$-th round in Algorithm 9 as $A_i$, and the element added in the $i$-th round as $a_i$. Since $\mathcal{P}''$ is a matroid, we have that for any $O' \in \mathcal{P}''$, there exists an ordering of $O' = \{o'_1, o'_2, ..., o'_t\}$ such that for each $i \in [t]$, $A_i/S \cup \{o'_i\} \in \mathcal{P}''$. Therefore, for each $i \in [t]$, $A_i \cup \{o'_i\} \in \mathcal{P}$. By the greedy selection rule in Algorithm 9, we have that

$$\Delta f(A_i, a_i) \geq \Delta f(A_i, o'_i) \geq \Delta f(A, o'_i),$$

where the second inequality follows from the fact that $A$ is the output of Algorithm 9 and that $A_i \subseteq A$. Summing over all $i$, we can get that

$$f(A) - f(S) \geq \sum_i \Delta f(A, o'_i) \geq \Delta f(A, O'). \tag{13}$$

Summing over (12) and (13), we can get that

$$f(A) - f(\emptyset) \geq \Delta f(A, O') + \Delta f(S, O)$$
$$\geq \Delta f(A, O') + \Delta f(A, O)$$
$$\geq \Delta f(A, O' \cup O).$$

Since the above inequality holds for any $O \in \mathcal{P}'$ and $O' \in \mathcal{P}''$. Therefore,

$$f(A) - f(\emptyset) \geq \max_{O \in \mathcal{P}', O' \in \mathcal{P}''} \Delta f(A, O' \cup O).$$

Notice that any set in $\mathcal{P}$ can be decomposed into the union of a set in $\mathcal{P}'$ and a set in $\mathcal{P}''$. It then follows that $\max_{O \in \mathcal{P}', O' \in \mathcal{P}''} \Delta f(A, O' \cup O) \geq \Delta f(A, OPT)$. Therefore, $f(A) \geq f(OPT)/2$. $\qquad \square$

## F  APPENDIX FOR SECTION 3

In this section, we present the additional experimental setup and results omitted in Section 3. In particular, we present additional details about the experimental setup in Section F.1, and additional experimental results in Section F.2.

### F.1  EXPERIMENTAL SETUP

In this section, we provide additional details about the applications used to evaluate our algorithms, which include set cover, max cover, and graph cut. Below, we define each application and describe the associated setup in detail.

In the application of set cover, the function $f$ is defined to be the number of tags covered by the elements in a subset. The problem is defined as follows.

**Definition F.1.** (**Set Cover**) Suppose there are a total of $n$ elements denoted as $U$. Let $T$ be a set of tags. Each element in $U$ is tagged with a set of elements from $T$ via a function $t : U \to 2^T$. The function $f$ is defined as

$$f(S) = |\cup_{s \in S} t(s)|, \qquad \forall S \in U.$$

Next, we introduce the definition of max cover, which is a monotone submodular function defined on graphs.

**Definition F.2.** (**Max Cut**) Let $G = (V, E)$ be a graph, and $w : E \to \mathbb{R}_{\geq 0}$ be a function that assigns a weight to every edge in the graph. The function $f : 2^V \to \mathbb{R}_{\geq 0}$ maps a subset of vertices $X \subseteq V$ to the total weight of edges between $X$ and $V \setminus X$. More specifically,

$$f(X) = \sum_{x \in X \text{ or } y \in X} w(x, y).$$

We also evaluate our experiments on the instance of image summarization. For this task, we use a subset of the ImageNet dataset (ImageNet_50).

**Definition F.3.** (**Image Summarization**) Let $N \subseteq \mathbb{R}^d$ denote the ground set, where each item $x \in N$ (e.g., an image) is represented by a feature vector. The objective is to maximize the *Determinantal Point Process (DPP)* function (Iyer and Bilmes, 2015), which is a monotone submodular function defined as:

$$f(S) = \log \det(I + K_S),$$

where $I$ is the identity matrix, $K \in \mathbb{R}^{|N| \times |N|}$ is a positive semidefinite kernel matrix, and $K_S$ denotes the principal submatrix of $K$ indexed by the subset $S \subseteq N$.

For general SCP, where $f$ can be nonmonotone, the application we consider is where $f$ is a graph cut function, which is a submodular but not necessarily monotone function.

**Definition F.4.** (**Graph Cut**) Let $G = (V, E)$ be a graph, and $w : E \to \mathbb{R}_{\geq 0}$ be a function that assigns a weight to every edge in the graph. The function $f : 2^V \to \mathbb{R}_{\geq 0}$ maps a subset of vertices $X \subseteq V$ to the total weight of edges between $X$ and $V \setminus X$. More specifically,

$$f(X) = \sum_{x \in X, y \in V \setminus X} w(x, y).$$

Next, we present more details about the experimental setup in the order of the problems we consider. For the experiments on nonmonotone SCP, the dataset is the email-Euall dataset, where the dataset is partitioned into 5 different subgroups based on the synthetic labels of the dataset. The group proportions are uniform, i.e., the parameter $p_j$ used in this experiment satisfies $p_1 = p_2 = \cdots = p_5 = 1/5$. To speed up the experiments, the conversion algorithm's subroutine is parallelized across 10 threads. Additional details about the values of the parameters in the experiments are presented as follows. The parameter $\alpha = 0.2$, $\epsilon = 0.05$ and $\delta = 0.1$.

For the experiments on monotone SCKP, we run the experiments on two instances, which include max cover and set cover. For the max cover instance, we use a subset of the Twitch Gamers dataset (Rozemberczki and Sarkar, 2021), selecting 2,000 users speaking six major languages which include English, German, French, Spanish, Russian, and Chinese. For the set cover instance, we use two datasets here. The first one is the core dataset, which is the Corel5k set of images in Duygulu et al. (2002) ($n = 4500$). We assign a label to each element in the dataset uniformly selected from $\{0, 1, 2, 3, 4\}$. Another dataset we use here is the synthetic dataset. The synthetic dataset is generated with 5 partitions with $40 * i + 200$ number of elements in partition $i$ for each $i \in [4]$. The synthetic dataset has a similar structure as the tightness example in Section E.4 in the appendix. In the first partition, each element is mapped to a disjoint set of tags. For the elements partition $i$ where $i > 1$, the mapped sets of tags of 100 elements are the same, with the other elements mapped to disjoint sets of 25 tags. The cost of each element in the synthetic dataset and in the twitch dataset is generated randomly in the range of $[0.001, 10]$. The other parameters in the experiments comparing different values of $\tau$ include: $\alpha = 0.2$, $\epsilon = 0.05$. The parameters for the experiments comparing different values of $\alpha$ include: threshold value $\tau = 700$, $\epsilon = 0.05$. The parameter $p_j$ used in this experiment satisfies $p_1 = p_2 = \cdots = p_5 = 1/5$. Next, we illustrate the two algorithms used in the experiments. The GREEDY algorithm uses the converting theorem in Algorithm 4 with the subroutine being a greedy algorithm. In particular, the subroutine greedy algorithm adds the element $s = \arg\max_{x:S\cup x\in\mathcal{P}} \frac{\Delta f(S,x)}{c(x)}$ to the solution set $S$, where $\mathcal{P} = \{S \subseteq U : c(S \cap U_j) \le p_j v, \forall j \in [N]\}$. Here the subroutine algorithm of GREEDY is not guaranteed with any approximation ratio. In this sense, this algorithm can be regarded as a heuristic algorithm. The GREEDY-Knapsack algorithm runs by iteratively adding the element with the highest density of marginal gain, i.e., $s = \arg\max_{x\in U} \frac{\Delta f(S,x)}{c(x)}$ until $f(S) \ge (1 - \epsilon)\tau$.

For the SCF experiments, we consider the same synthetic dataset and the corel dataset used in the experiment for SCKP. Apart from these datasets, we also consider the image summarization task, where the goal is to select a diverse and representative image subset across all classes. The dataset used here is ImageNet (Deng et al., 2009), consisting of 50 classes and 26,112 images (ImageNet_50). Each image is represented by a feature vector extracted using ResNet-18. Additionally, in our experiment, we set $K$ as a Gaussian kernel matrix such that $K_{ij} = e^{-||x_i - x_j||^2/\sigma^2}$.

To ensure a fair comparison among the used algorithms, we keep the approximation ratio on the function value $f$ the same by setting $\varepsilon = 0.05$ for THRES-Fair and $\varepsilon = 0.1$ for GREEDY-Fair and BLOCK-G-Fair while keeping the other parameters the same.

### F.2 ADDITIONAL EXPERIMENTAL RESULTS

The additional experimental results comparing different algorithms for different problems are presented in Figure 2, 3 and 4. The additional experimental results for SCKP algorithms on the corel dataset and the synthetic dataset are presented in Figure 2 and 3. The results demonstrate that our algorithm, BLOCK-G, achieves a slightly lower cost compared to GREEDY and significantly outperforms GREEDY-Knapsack in this regard. Additionally, BLOCK-G requires substantially fewer function queries and has a much faster runtime than GREEDY, highlighting its practical efficiency. The function values $f$ for different algorithms are similar, which is because all algorithms (including baselines) in the experiments are designed to terminate once the function value reaches $(1 - \epsilon)\tau$. For the experiments on different values of $\alpha$, we can see that increasing $\alpha$ significantly reduces the number of evaluations, leading to dramatic improvements in runtime. This also corresponds to the results of query complexity of our converting algorithm (Algorithm 5) as proved in Theorem C.3.

Further results on the query complexity and runtime for the non-monotone SCP problem are provided in Figures 4(i) and 4(j). From the results, we can see that the BLOCK-G algorithm runs faster than

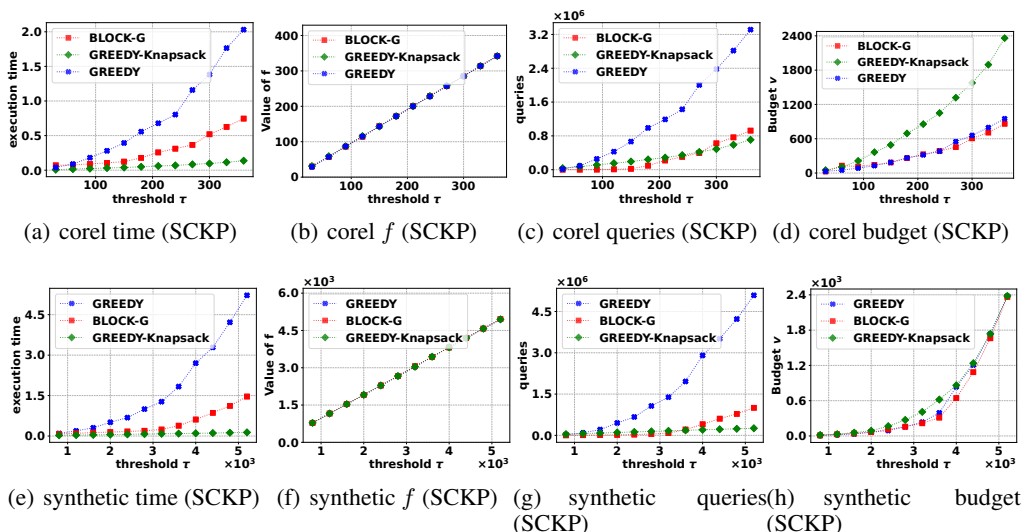

Figure 2: The experimental results of running the algorithms on the Corel5k dataset and the synthetic dataset. Samples: the number of queries. Budget: $\max_{i \in [N]} \frac{c(S \cap U_i)}{p_i}$.

the STREAM algorithm and the GUIDED-RG algorithm, which demonstrates the efficiency of our algorithm.

The additional results of comparing different algorithms in terms of the query complexity for the experiments on the SCF problem are presented in Figure 4(c) and 4(g). From the results, we can see that the query complexity of the BLOCK-G-Fair algorithm is better than that of the GREEDY-Fair algorithm, and is worse than THRES-Fair. This is because these three algorithms differ in the subroutine algorithm used in the converting algorithm in Algorithm 1 in Chen et al. (2025) developed to convert an algorithm for SMF to SCF. Specifically, THRES-Fair used the threshold greedy algorithm, which runs in time complexity of $\mathcal{O}(\frac{n}{\epsilon} \log \frac{n}{\epsilon})$ while the subroutine algorithms for BLOCK-G-Fair and GREEDY-Fair both run in time $\mathcal{O}(nk_g\beta)$ where $k_g$ is the guess of $|OPT|$ and the parameter $\beta$ refers to the approximation ratio. Therefore, the query complexity of BLOCK-G-Fair is lower than GREEDY-Fair because the parameter $\beta$ for BLOCK-G-Fair is $\frac{\ln \frac{1}{\epsilon}}{\ln 2}$, which is smaller than the GREEDY-Fair, which is $\mathcal{O}(\frac{1}{\epsilon})$.

The results of the function values for different assignments of $\tau$ on the experiments of SCF are presented in Figure 4(f) and 4(b). From the plots, we can see that the function values of the returned solutions of different algorithms are almost the same, and are linear in the threshold value $\tau$. This aligns with our theoretical guarantee of different algorithms, which requires that $f(S) \geq 0.9\tau$ for all of the algorithms. Finally, we also provide the results of the execution time of running different algorithms in Figure 4(d), and 4(h).

The additional experimental results on the ImageNet_50 dataset are presented in Figure 5. From these results, we observe that block-greedy consistently achieves significantly better fairness performance and, in many cases, returns solutions with lower or comparable cost to baselines. This demonstrates its practical effectiveness, especially in fairness-sensitive applications.

Finally, we also plot the distribution of different labels in the solutions produced by these algorithms on the corel dataset with $\tau = 300$, as is presented in Figure 6(a), 6(b), and 6(c). From the plots, we can see that over $30\%$ of the elements in the solution returned by GREEDY-Fair and THRES-Fair have the label 1, which indicates a lack of fairness in the output distribution. While the solutions produced by our algorithm BLOCK-G-Fair exhibit significantly fairer distributions across different labels, demonstrating the effectiveness of our proposed algorithms.

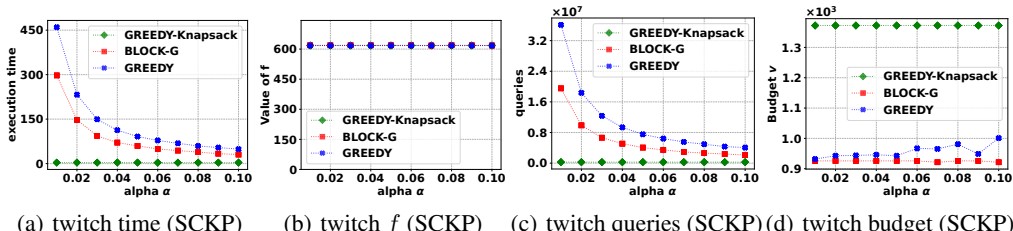

(a) twitch time (SCKP)    (b) twitch $f$ (SCKP)    (c) twitch queries (SCKP) (d) twitch budget (SCKP)

Figure 3: The experimental results for the SCKP problem on the Twitch dataset across different $\alpha$ values. Samples: the number of queries. Budget: $\max_{i \in [N]} \frac{c(S \cap U_i)}{p_i}$.

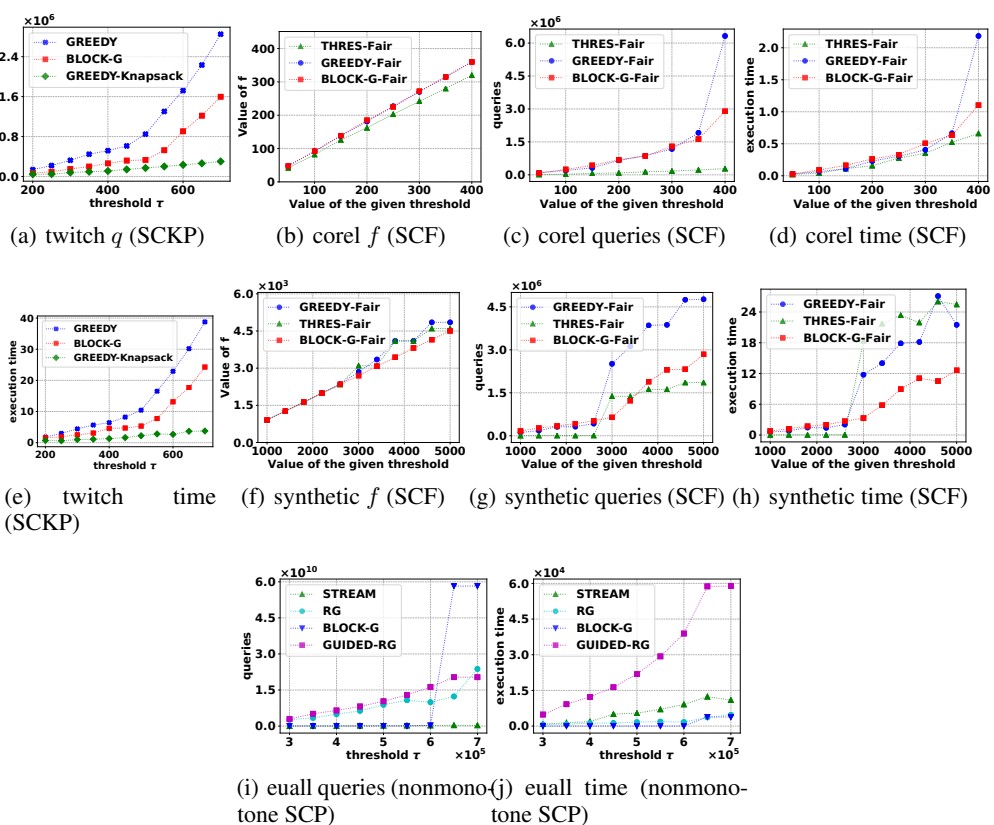

(a) twitch $q$ (SCKP)    (b) corel $f$ (SCF)    (c) corel queries (SCF)    (d) corel time (SCF)

(e) twitch time (SCKP)    (f) synthetic $f$ (SCF)    (g) synthetic queries (SCF) (h) synthetic time (SCF)

(i) euall queries (nonmono-(j) euall time (nonmono-tone SCP)    tone SCP)

Figure 4: The experimental results of running the algorithms on the Corel5k dataset and the synthetic dataset. Samples: the number of queries. Cost: the size of the returned solution. Budget: $\max_{i \in [N]} \frac{c(S \cap U_i)}{p_i}$. Fairness difference: $(\max_c |S \cap U_c| - \min_c |S \cap U_c|)/|S|$.

## G  BROADER IMPACT

This paper presents work whose goal is to advance the field of Machine Learning. There are many potential societal consequences of our work, none of which we feel must be specifically highlighted here.

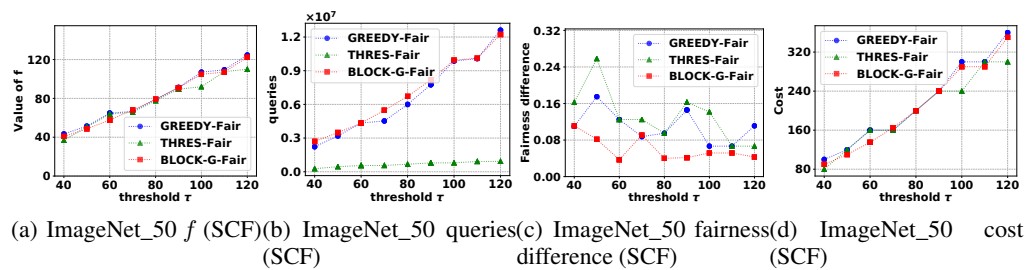

(a) ImageNet_50 $f$ (SCF)(b)  ImageNet_50 queries(c) ImageNet_50 fairness(d)  ImageNet_50  cost
(SCF)                              difference (SCF)       (SCF)

Figure 5: The experimental results of running the algorithms on the ImageNet_50 dataset on the SCF problem. Samples: the number of queries. Cost: the size of the returned solution. Fairness difference: $(\max_c |S \cap U_c| - \min_c |S \cap U_c|)/|S|$.

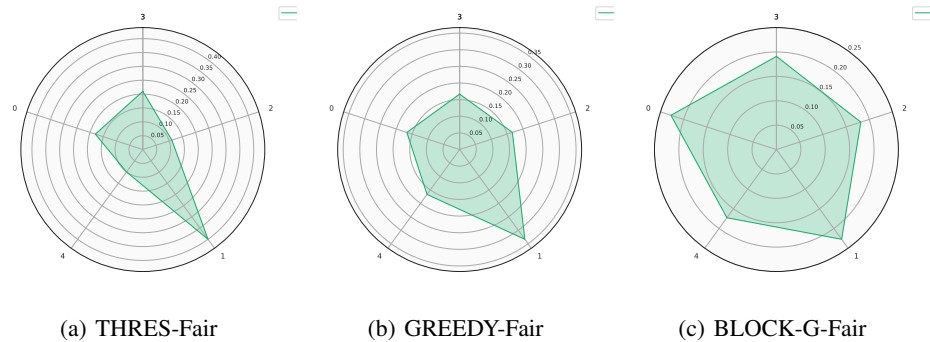

(a) THRES-Fair            (b) GREEDY-Fair            (c) BLOCK-G-Fair

Figure 6: Radar plots of the label distributions for the experiments on the instance of SCF on the corel dataset with $\tau = 300$.

