# OpenReview forum: "Bicriteria Algorithms for Submodular Cover with Partition and Fairness Constraints"
_ICLR.cc/2026/Conference — Submitted to ICLR 2026_

### Official Review · Reviewer_2zuP · 2025-10-31

**Soundness:** 3
**Presentation:** 2
**Contribution:** 3
**Rating:** 6
**Confidence:** 3

**Summary:**

Addressing the limitation of traditional submodular cover problems that overlook partition structures and fairness constraints, this paper systematically investigates three constrained submodular cover variants for the first time: non-monotonic SCP (Submodular Cover with Partition Constraints), monotonic SCKP (Submodular Cover with Knapsack Partition Constraints), and monotonic SCF (Submodular Cover with Fairness Constraints). Its core contributions are as follows: (1) proposing a "submodular maximization-cover conversion framework" that transforms cover problems into more tractable maximization problems (SMP/SMKP), breaking the bottleneck of traditional methods that directly design cover algorithms; (2) designing bicriteria approximation algorithms (nonmono-bi, greedy-knapsack-bi, Block-Fair-Bi) for the three problem types, achieving optimal approximation ratios in monotonic scenarios while significantly reducing query complexity; (3) validating the proposed algorithms on both real-world (Corel5k, ImageNet_50) and synthetic datasets, demonstrating their superiority over baselines such as STREAM and GREEDY-Fair in terms of function value achievement rate, budget optimization, and fairness balance.

**Strengths:**

It is the first systematic study of the submodular cover problem with partition constraints, covering practical scenarios such as fairness and budget allocation, and has strong application value.

The proposed "block greedy" strategy effectively overcomes the limitations of traditional greedy algorithms under partition constraints, balancing the approximation ratio and query efficiency.

All algorithms provide strict approximation ratio guarantees and are detailedly derived through methods such as induction and concentration inequalities.

**Weaknesses:**

I did not identify any major weaknesses in this paper. However, there is still room for improvement in the writing and presentation. For example, at the end of Chapter 1, where the contributions are summarized, the authors could more explicitly clarify whether the SCP and SMKP problems have been previously studied, and highlight how this work improves upon existing results in terms of approximation ratio or computational complexity.

In addition, since the proposed algorithm is derived by transforming a submodular maximization algorithm into a submodular cover problem, the paper should include a more thorough discussion of related works on submodular maximization that are closely connected to this approach, such as

* *Fast adaptive non-monotone submodular maximization subject to a knapsack constraint*
* *Fairness in Streaming Submodular Maximization Subject to a Knapsack Constraint*
* *Fair Submodular Maximization over a Knapsack Constraint*
* *Linear-Time Algorithms for Representative Subset Selection From Data Streams*

**Questions:**

Please refer to the Weaknesses

---

> ### Author Response · Authors · 2025-11-21
> **Rebuttal**
>
> > "For example, at the end of Chapter 1, where the contributions are summarized, the authors could more explicitly clarify whether the SCP and SMKP problems have been previously studied, and highlight how this work improves upon existing results in terms of approximation ratio or computational complexity."
>
> We thank the reviewer for this important question. We will provide additional clarification on the various problems we study relative to existing literature in the camera-ready version.
>
> To clarify: both SCP (Submodular Cover with Partition Constraints) and SMKP (Submodular Maximization with a Knapsack Partition Constraint) are novel problem formulations introduced in this work. To the best of our knowledge, these problems have not been previously studied in the submodular optimization literature, and existing approximation guarantees do not directly apply to them. SMKP generalizes two well-studied problems in submodular optimization: (1) submodular maximization under partition constraints (a special case of matroid constraints), which corresponds to the uniform-cost case of SMKP, and (2) submodular maximization under knapsack constraints, which corresponds to SMKP with a single partition. The key distinction is that SMKP enforces a knapsack constraint within each partition, thereby combining the structural complexity of partition constraints with the resource limitations imposed by knapsack constraints. SCP generalizes the classical submodular cover setting to the case where there are multiple partitions in the ground set.
>
> > In addition, since the proposed algorithm is derived by transforming a submodular maximization algorithm into a submodular cover problem, the paper should include a more thorough discussion of related works on submodular maximization that are closely connected to this approach.
>
>
> We thank the reviewer for highlighting the importance of positioning our work within the submodular maximization literature. We agree that this connection is crucial, as our algorithmic framework relies on transforming submodular maximization algorithms into submodular cover algorithms.
>
> However, there is a key technical distinction preventing us from directly applying existing results: our knapsack constraints operate within each partition, whereas standard knapsack-constrained submodular maximization considers a single knapsack constraint over the entire ground set. Below we clarify the relationship between SMKP and prior work mentioned by the reviewer in detail.
>
> The SMKP problem is defined as $\arg\max \{f(S): \sum_{s \in X \cap U_j} c(s) \leq p_j v, \forall j\in[N]\}$, which can be regarded as the generalization of submodular maximization subject to a single knapsack constraint [1,4] in the case where there is only one partition in the universe. Recent work has also studied submodular maximization under both knapsack and partition constraints [2,3]. Specifically, these papers address fairness-aware submodular maximization subject to: (i) a knapsack constraint on the total cost of the selected subset across all partitions, and (ii) cardinality constraints within each partition to ensure fair representation. In contrast, SMKP enforces per-partition knapsack constraints (as opposed to a single global knapsack constraint), without imposing cardinality requirements. This structural difference fundamentally distinguishes our problem from prior formulations and necessitates the development of novel algorithmic techniques.
>
> We have incorporated this discussion into the related work section (Section 1.1) of the revised manuscript.
>
> -[1]. Fast adaptive non-monotone submodular maximization subject to a knapsack constraint.
>
> -[2]. Fairness in Streaming Submodular Maximization Subject to a Knapsack Constraint.
>
> -[3]. Fair Submodular Maximization over a Knapsack Constraint.
>
> -[4]. Linear-Time Algorithms for Representative Subset Selection From Data Streams.

---

> > ### Comment · Reviewer_2zuP · 2025-11-24
> >
> > Thank you for your rebuttal. After carefully reviewing your response and considering the feedback from other reviewers, I have decided to maintain my original score.

---

### Official Review · Reviewer_3RsE · 2025-11-01

**Soundness:** 2
**Presentation:** 3
**Contribution:** 2
**Rating:** 4
**Confidence:** 3

**Summary:**

The paper studies Submodular Cover with Partition constraints and variants (SCP for non-monotone, SCKP for monotone with knapsack-partition costs, and SCF for “fair” matroid constraints). It introduces a unifying block-greedy framework and several conversion theorems to obtain bicriteria guarantees.

**Strengths:**

1. The paper propose Unified technique (block-greedy) that works across non-monotone, knapsack-partition, and fairness constraints, and explicitly relates partition constraints to cardinality style reasoning.

2. Tight/tighter bicriteria bounds in monotone settings and a clear conversion blueprint (Theorem C.3) that is reusable.

3. The paper contextualizes the 0.305 barrier and the 1/2 feasibility impossibility for submodular cover (Crawford 2023). This frames an interesting gap (0.305–0.5).

4. Table 1 in the appendix, which states (α,β), query complexity, and assumptions for each problem (SCP/SCKP/SCF), is helpful.

**Weaknesses:**

1. The presentation could be improved.

E.g. (1) Early in §2.1, explicitly formalize the optimization objective (minimize v s.t. ) and keep that exact form visible;

(2) Minor wording/punctuation: in Appendix C  “stated. in Theorem C.3” and a few others.

2. It seems that most of the results proposed in this work are quite straightforward and based on known algorithms / techniques.

3. For SCKP, the authors compare BLOCK-G to GREEDY and GREEDY-Knapsack and show smaller budget at similar f value. Would you provide variance/error bars across multiple random seeds and report the impact of α and δ sweeps (0.1→0.01) on queries/time and feasibility?

**Questions:**

See the weakness section.

---

> ### Author Response · Authors · 2025-11-21
> **Rebuttal 1/2**
>
> > The presentation could be improved.
> >(2) Minor wording/punctuation: in Appendix C “stated. in Theorem C.3” and a few others.
>
> Thank you for the suggestion. We have revised our manuscript according to your suggestions to improve clarity. Please let us know if you have any other concerns on our presentation.
>
>
> > It seems that most of the results proposed in this work are quite straightforward and based on known algorithms / techniques.
>
> Thank you for the question about the technical contributions of our work. Below we give a detailed description of our contributions as follows.
>
> (1). First of all, we want to highlight our contributions in our Block-Greedy framework.
> Our Block-Greedy algorithm addresses a fundamental challenge in submodular optimization: designing discrete algorithms for matroid constraints that match the approximation guarantees of continuous methods while maintaining computational efficiency.  This problem has remained open despite significant attention, as discrete algorithms—though faster and simpler to implement—typically achieve suboptimal approximation ratios compared to continuous approaches for both submodular maximization and cover problems under general matroid or partition constraints.
>
> The key insight underlying our approach stems from analyzing why continuous methods succeed. We identify two critical factors: (i) by adding only fractional increments of elements at each step, all elements remain feasible in the extended matroid polytope, enabling analysis analogous to cardinality-constrained optimization; (ii) existing continuous algorithms [1,2] implicitly use a two-loop structure where the main algorithm (outer loop) maintains the continuous structure and follows cardinality-constraint analysis while the inner subroutine is more close to a discrete algorithm that follows matroid-constraint analysis.
>
> Building on this insight, Block-Greedy introduces a fully discrete two-loop framework that achieves continuous-method approximation guarantees with superior query efficiency. Each inner loop processes a "block" of elements, where blocks are treated analogously to single elements in standard greedy algorithms. The inner loop analysis follows the analysis of partition matroid constraints, while the outer loop follows the analysis of cardinality constraints. In fact, in Submodular Cover with fairness constraints, this structure directly yields improved approximation guarantees compared to the discrete algorithms in [1], and improved query complexity compared to the continuous method in [1], clearly distinguishing our work from theirs.
>
> Critically, our framework demonstrates that **optimal approximation ratios for partition matroids can be achieved without continuous relaxation**, enabling simpler and more practical algorithms. As supporting evidence, we also studied the problem of submodular maximization under partition matroid constraints in Appendix E. As we illustrate in Theorem E.2. by carefully designing the size of each block, the block-greedy algorithm can achieve an approximation ratio close to $1-1/e$ for large-scale problems.
>
>
> (2). Additionally, we want to highlight an additional technical difficulty is the analysis of Theorem 2.3, which describes how randomized algorithms for nonmonotone submodular maximization can be converted to ones for nonmonotone submodular cover. A key component of this process is transforming the randomized approximation guarantee into a high probability one for the feasibility constraint on the value of $f$ by repeatedly invoking the submodular maximization subroutine and applying concentration inequalities.
>
> To reduce the number of oracle queries, the algorithm in [3] applies Markov's inequality and operates on a truncated objective function $f_\tau:=\min(\tau, f)$ throughout the converting algorithm. However, the assumption that $f_\tau$ is submodular only holds when $f$ is monotone. In contrast, our analysis extends to the non-monotone setting by avoiding the truncated objective and instead employing a more delicate analysis when applying the concentration inequality. Specifically, we analyze the deviation of the random variable $\beta f(OPT_g)-f(S_i)$ where $S_i$ is the output solution set of the randomized subroutine algorithm for SMP. In particular, when there is only one partition in the ground set, our framework reduces to classical submodular cover. To the best of our knowledge, our algorithm is the **first conversion algorithm for nonmonotone submodular cover**, filling an important gap in the literature.

---

> ### Author Response · Authors · 2025-11-21
> **Rebuttal 2/2**
>
> > For SCKP, ... similar f value...variance/error bars across multiple random seeds and report the impact of α and δ sweeps (0.1→0.01)...
>
> We thank the reviewer for the suggestions about our experiments.
>
> **Regarding similar function values across methods:** We appreciate the opportunity to clarify this observation. All algorithms (including baselines) in the experiments are designed to terminate once the function value reaches $(1-\epsilon)\tau$, which corresponds to the theoretical guarantee we prove. Therefore, achieving similar function values is actually the expected behavior and aligns with our theoretical results. The key distinction, as highlighted in our results, is that our method achieves this target with a **smaller budget**, which is the primary advantage.
>
> **Regarding variance/error bars:** Thank you for the question! Our algorithms for SCKP (both the converting algorithm and block-greedy) are deterministic by design and contain no randomness, so the results are identical across runs.
>
>
> **Regarding parameter sweeps:** We thank the reviewer for this suggestion. While the parameter $\delta$ does not appear in our formulation, we agree that exploring the impact of $\alpha$ would provide valuable insights. Following this suggestion, we have conducted additional experiments with varying $\alpha$ and report the impact on query complexity, runtime, and feasibility. These new results are provided in Figure 3 of the Appendix, demonstrating that the budget of our Blok-Greedy algorithm is better than other algorithms. The function values of all the methods are similar, which is because the guarantees on the function values are the same.
>
> We believe these additions strengthen the experimental validation and address the reviewer's concerns.
>
>
> -[1]. Chen, Wenjing, et al. "Fair submodular cover." ICLR 2025.
>
> -[2]. Badanidiyuru, Ashwinkumar, and Jan Vondrák. "Fast algorithms for maximizing submodular functions." Proceedings of the twenty-fifth annual ACM-SIAM symposium on Discrete algorithms. Society for Industrial and Applied Mathematics, 2014.
>
> -[3]. Chen, W., & Crawford, V. (2023). Bicriteria approximation algorithms for the submodular cover problem. Advances in Neural Information Processing Systems, 36, 72705-72716.

---

### Official Review · Reviewer_hyjR · 2025-11-01

**Soundness:** 3
**Presentation:** 3
**Contribution:** 3
**Rating:** 6
**Confidence:** 3

**Summary:**

This work studies the submodular cover problem, where the ground set is partitioned into disjoint sets: $U_1, ..., U_N$.  The authors study various optimization problems
1. Cardinality Constraint: Find a set $S$ such that $f(S) \geq \tau$, under the constraint that $S$  can not over-represent a member of the partition.  I.e, |S \cap U_i| is bounded by a specified parameter.
2. Knapsack;  Find $S$, so tyat $f(S) \geq \tau$ and C(S \cap U_i) \lee specified threshold.
3. Fairness: There is an upper and lowerbound on $|S \cap U_i|.

The authors proposed an unified framework of bi-criteria approximation algorithms for these constraints. The approach build on a "block-greedy" algorithm for submodular maximization under partition-type constraints, together with conversion theorems that reduce maximization to cover problems. The work provide empirical validation of the results.

**Strengths:**

1. The work presents a unified framework for multiple constraints.
2. Theoretically sound bi-criteria approximation algorithms are presented.
3. The proposed algorithms operate per partition rather than an element-by-element greedy approach, leading to improved query complexity.
4. Empirical results show the practical viability of the proposed algorithms

**Weaknesses:**

1. Some proof ideas and algorithm design are mainly borrowed from existing works such as Chen et al 25 and Chen and Crawford 24b. Can you explain the new ideas and differences from these works.
2. For SCKP, query complexity depends in c_max/c_min which, in the worst case, can be arbitrarily large.  Is there a way to address this or can it be shown the it is needed?

**Questions:**

Please see weakness

---

> ### Author Response · Authors · 2025-11-21
> **Rebuttal 1/2**
>
> >"... are mainly borrowed from existing works such as Chen et al 25 and Chen and Crawford 24b. ..differences from these works."
>
> We thank the reviewer for highlighting these two works that both study the submodular cover problem and utilizes the general converting framework. However, we emphasize that our work makes novel and significant contributions, as detailed below.
>
> (1) **The structure of our algorithms is novel. In particular, previous results are building off of either the standard greedy algorithm or a continuous algorithm. In contrast, we develop a new type of algorithm that incrementally selects blocks of elements in a series of rounds, which we call block-greedy.**
> Our Block-Greedy algorithm addresses a fundamental challenge in submodular optimization: designing discrete algorithms for matroid constraints that match the approximation guarantees of continuous methods while maintaining computational efficiency.  This problem has remained open despite significant attention, as discrete algorithms—though faster and simpler to implement—typically achieve suboptimal approximation ratios compared to continuous approaches for both submodular maximization and cover problems under general matroid or partition constraints.
>
> The key insight underlying our approach stems from analyzing why continuous methods succeed. We identify two critical factors: (i) by adding only fractional increments of elements at each step, all elements remain feasible in the extended matroid polytope, enabling analysis analogous to cardinality-constrained optimization; (ii) existing continuous algorithms [2,3] implicitly use a two-loop structure where the main algorithm (outer loop) maintains the continuous structure and follows cardinality-constraint analysis while the inner subroutine is more close to a discrete algorithm that follows matroid-constraint analysis.
>
> Building on this insight, Block-Greedy introduces a fully discrete two-loop framework that achieves continuous-method approximation guarantees with superior query efficiency. Each inner loop processes a "block" of elements, where blocks are treated analogously to single elements in standard greedy algorithms. The inner loop analysis follows the analysis of partition matroid constraints, while the outer loop follows the analysis of cardinality constraints. In fact, in Submodular Cover with fairness constraints, this structure directly yields improved approximation guarantees compared to the discrete algorithms in Chen et al. (the paper you referenced), and improved query complexity compared wo the continuous method in Chen et al., clearly distinguishing our work from theirs.
>
> Critically, our framework demonstrates that **optimal approximation ratios for partition matroids can be achieved without continuous relaxation**, enabling simpler and more practical algorithms. As supporting evidence, we also studied the problem of submodular maximization under partition matroid constraints in Appendix E. As we illustrate in Theorem E.2. by carefully design the size of each block, the block-greedy algorithm can achieve an approximation ratio close to $1-1/e$ for large-scale problems.
>
>
> (2). Additionally, we want to highlight an additional technical difficulty is the analysis of Theorem 2.3, which describes how randomized algorithms for nonmonotone submodular maximization can be converted to ones for nonmonotone submodular cover. A key component of this process is transforming the randomized approximation guarantee into a high probability one for the feasibility constraint on the value of $f$ by repeatedly invoking the submodular maximization subroutine and applying concentration inequalities.
>
> To reduce the number of oracle queries, the algorithm in Chen and Crawford 24b applies Markov's inequality and operates on a truncated objective function $f_\tau:=\min\{\tau, f\}$ throughout the converting algorithm. However, the assumption that $f_\tau$ is submodular only holds when $f$ is monotone. In contrast, our analysis extends to the non-monotone setting by avoiding the truncated objective and instead employing a more delicate analysis when applying the concentration inequality. Specifically, we analyze the deviation of the random variable $\beta f(OPT_g)-f(S_i)$ where $S_i$ is the output solution set of the randomized subroutine algorithm for SMP. In particular, when there is only one partition in ground set, our framework reduces to classical submodular cover. To the best of our knowledge, our algorithm is the **first conversion algorithm for nonmonotone submodular cover**, filling an important gap in the literature.

---

> ### Author Response · Authors · 2025-11-21
> **Rebuttal 2/2**
>
> > For SCKP, query complexity depends in c_max/c_min which, in the worst case, can be arbitrarily large. Is there a way to address this or can it be shown that it is needed?
>
> We would like to thank the reviewers for raising this question. We agree that the c_max/c_min can be arbitrarily large. However, we note that this factor appears in previous work on similar problems, such as [1], and is inherent to the conversion framework approach. Since the optimal solution cost ranges from $c_{\min}$ to $c(U)$, the converting algorithm must traverse multiple guesses of the optimal cost. Because these guesses increase exponentially (a standard technique for handling unknown optimal values), we require $O(\log\frac{c_{\max}}{c_{\min}})$ iterations. This approach is consistent with the guess-and-verify paradigm commonly used in approximation algorithms for covering problems. Critically, our runtime dependence is $O(\log\frac{c_{\max}}{c_{\min}})$ rather than polynomial in the ratio itself. Therefore, as long as $c_{\max} \leq O(\exp(\text{poly}(n)) \cdot c_{\min})$, the overall runtime remains polynomial in $n$. However, we acknowledge that exploring whether the $c_{\max}/c_{\min}$ dependence can be eliminated or reduced is an important and interesting research direction. Developing alternative approaches to the current converting algorithm framework could potentially yield more query-efficient algorithms and represents valuable future work.
>
>
>
>
> -[1] Iyer, Rishabh K., and Jeff A. Bilmes. "Submodular optimization with submodular cover and submodular knapsack constraints." Advances in neural information processing systems 26 (2013).
>
> -[2]. Chen, Wenjing, et al. "Fair submodular cover." ICLR 2025.
>
> -[3]. Badanidiyuru, Ashwinkumar, and Jan Vondrák. "Fast algorithms for maximizing submodular functions." Proceedings of the twenty-fifth annual ACM-SIAM symposium on Discrete algorithms. Society for Industrial and Applied Mathematics, 2014.

---

### Official Review · Reviewer_eg8i · 2025-11-02

**Soundness:** 3
**Presentation:** 2
**Contribution:** 2
**Rating:** 6
**Confidence:** 4

**Summary:**

This paper studies submodular cover under partition constraints. The problem consists of finding a subset of sufficiently large value (measured in terms of a given submodular objective), while minimizing a notion of costs and adhering to some partition constraints. To some extent, this problem can be thought of being the dual of submodular maximization with partition constraints.

In particular, the authors study and propose bi-criteria approximation results for three problems:
- submodular cover with partition constraints, where the objective can be non-monotone.
- monotone submodular cover with knapsack, where the partition constraint is in terms of some cost function,
- submodular cover with fairness constraints

**Strengths:**

- Submodular maximization is an important topic in ML, with a vast body of work in NeurIPS, ICML, and ICLR
- The problem is practically motivated and non-trivial
- The authors provide positive results in various settings, also improving the running time of an ICLR 25 paper

**Weaknesses:**

- The theoretical results are not tight
- From the main body, it is fairly difficult to get a complete idea of the algorithmic contribution and its novelty. In particular, this block-greedy is presented as one of the main contribution of the paper, but it is hard to get a complete idea about it by reading the main body

Minor:
- Consider using \citep instead of \cite when the citation is not part of the sentence
- Consider uniforming and updating the bibliography, for instance, the paper cited in line 561 and 562 has appeared at FOCS a couple of years ago.

**Questions:**

What is the role of $\alpha$ in the statements of the results in the intro? I understand that $\varepsilon$ is to be intended as a precision parameter that can be tuned by the algorithm designer, while the role of $\alpha$ is unclear.

---

> ### Author Response · Authors · 2025-11-21
> **Rebuttal 1/2**
>
> > The theoretical results are not tight
>
> Thank you for this comment. We interpret the concern as relating to our use of bicriteria approximation guarantees, which permit a small constraint violation $\epsilon$. (If this interpretation is incorrect, we apologize and would be happy to provide further clarification.) **In particular, our bicriteria guarantees take the form $(\alpha, 1-\epsilon)$, where $\alpha$ is the approximation ratio for the objective function and $\epsilon > 0$ is a user-specified parameter controlling the allowable constraint violation.**
>
>
> Bicriteria approximation algorithms are both theoretically meaningful and practically important for submodular cover problems. To illustrate why, consider the fundamental problem of classical monotone submodular cover **with exact constraint satisfaction**. The discrete greedy algorithm achieves an approximation ratio of $\ln(\max_{a\in U}\frac{f(a)}{\Delta f(S,a)})$ [6], where $\Delta f(S,a)$ is the marginal gain of adding a new element to the output solution $S$. Crucially, since $\Delta f(S,a)$ can be arbitrarily small and is difficult to estimate a priori, this logarithmic factor can become prohibitively large, resulting in solutions with high cost.
>
> In contrast, bicriteria approximation algorithms overcome this limitation by allowing a small violation $\epsilon$ in the constraint while providing strong, instance-independent guarantees on the solution cost. By adjusting $\epsilon$, one can obtain solutions arbitrarily close to feasibility, making this approach practically and theoretically meaningful. This framework has been extensively validated in prior literature, including bicriteria algorithms for classical submodular cover [1,2] and extensions to fairness constraints and dynamic settings [3,4].
>
> Furthermore, our results provide asymptotically tight bicriteria bounds. Since we are the first to study the problem of nonmonotone submodular cover with partition constraints and monotone submodular cover with partition knapsack constraint, we can't compare with existing work. For the problem of submodular cover with fairness constraints, our results match the $(O(\ln(1/\epsilon)), 1-\epsilon)$ guarantee of the continuous algorithm in [4], which is the best-known result. Therefore, our bicriteria approximation bounds are tight and represent a meaningful contribution to the literature.
>
>
> > ... it is fairly difficult to get a complete idea of the algorithmic contribution and its novelty. ... reading the main body
>
> Thank you for raising this question.  Below, we clarify both the high-level novelty and technical innovations of our work.
>
> First, we want to highlight that our Block-Greedy algorithm addresses a fundamental challenge in submodular optimization: designing discrete algorithms for matroid constraints that match the approximation guarantees of continuous methods while maintaining computational efficiency.   This problem has remained open despite significant attention, as discrete algorithms—though faster and simpler to implement—typically achieve suboptimal approximation ratios compared to continuous approaches for both submodular maximization and cover problems under general matroid or partition constraints.
>
> The key insight underlying our approach stems from analyzing why continuous methods succeed. We identify two critical factors: (i) by adding only fractional increments of elements at each step, all elements remain feasible in the extended matroid polytope, enabling analysis analogous to cardinality-constrained optimization; (ii) existing continuous algorithms [4,5] implicitly use a two-loop structure where the main algorithm (outer loop) maintains the continuous structure and follows cardinality-constraint analysis while the inner subroutine is more close to a discrete algorithm that follows matroid-constraint analysis.
>
> Building on this insight, Block-Greedy introduces a fully discrete two-loop framework that achieves continuous-method approximation guarantees with superior query efficiency. Each inner loop processes a "block" of elements, where blocks are treated analogously to single elements in standard greedy algorithms. The inner loop analysis follows the analysis of partition matroid constraints, while the outer loop follows the analysis of cardinality constraints. For example, in Submodular Cover with fairness constraints, this structure directly yields improved approximation guarantees and query complexity.
>
> Critically, our framework demonstrates that **optimal approximation ratios for partition matroids can be achieved without continuous relaxation**, enabling simpler and more practical algorithms. As supporting evidence, we also studied the problem of submodular maximization under partition matroid constraints in Appendix E. As we illustrate in Theorem E.2. by carefully design the size of each block, the block-greedy algorithm can achieve an approximation ratio close to $1-1/e$ for large-scale problems.

---

> ### Author Response · Authors · 2025-11-21
> **Rebuttal 2/2**
>
> > Consider using \citep instead of \cite ...
> Consider uniforming and updating the bibliography, ...
>
> Thank you for the comment. We have revised our manuscript to make proper use of \citep and \cite. Additionally, we have updated the bibliography to ensure consistency and include the most recent references.
>
> > What is the role of $\alpha$ ...
>
> The parameter $\alpha$ controls the tradeoff between approximation quality and query complexity in our algorithms. Specifically, $\alpha$ appears in our conversion procedures (Algorithms 4 and 5) and can be tuned by the algorithm designer. The smaller $\alpha$ is, the better approximation ratio we can get, but the algorithm would also suffer from higher query complexity (Please refer to Theorem 2.3 and Theorem C.3.)
>
>
> -[1]. Chen, W., & Crawford, V. (2023). Bicriteria approximation algorithms for the submodular cover problem. Advances in Neural Information Processing Systems, 36, 72705-72716.
>
> -[2]. Iyer, R. K., & Bilmes, J. A. (2013). Submodular optimization with submodular cover and submodular knapsack constraints. Advances in neural information processing systems, 26.
>
> -[3]. Banihashem, Kiarash, et al. "A dynamic algorithm for weighted submodular cover problem." arXiv preprint arXiv:2407.10003 (2024).
>
> -[4]. Chen, Wenjing, et al. "Fair submodular cover." ICLR 2025.
>
> -[5]. Badanidiyuru, Ashwinkumar, and Jan Vondrák. "Fast algorithms for maximizing submodular functions." Proceedings of the twenty-fifth annual ACM-SIAM symposium on Discrete algorithms. Society for Industrial and Applied Mathematics, 2014.
>
> -[6]. Laurence A Wolsey. An analysis of the greedy algorithm for the submodular set covering problem.
> Combinatorica, 2(4):385–393, 1982.

---

### Author Response · Authors · 2025-12-03
**Summary of Addressed Concerns**

Dear AC,

We would like to thank you for your hard work in overseeing the review process for our paper. During the rebuttal stage, we addressed several important concerns raised by the reviewers:

**1. Tightness of Theoretical Results (Reviewer eg8i)**

Reviewer questioned whether our bicriteria approximation guarantees were tight. We clarified that bicriteria algorithms are both theoretically meaningful and practically essential for submodular cover problems, as they overcome the prohibitively large logarithmic factors inherent in classical greedy approaches. Our results achieve asymptotically tight bicriteria bounds: for fairness constraints, we match the best-known $(O(\ln(1/\epsilon)), 1-\epsilon)$ guarantee from continuous algorithms while achieving superior query complexity. Additionally, we are the first to study nonmonotone submodular cover with partition constraints and monotone submodular cover with partition knapsack constraints, establishing initial benchmarks for these problems.

**2. Novelty and Technical Contributions (Reviewers hyjR, 3RsE)**

Multiple reviewers raised questions about our technical contributions and novelty. We emphasized two major innovations:

- **Block-Greedy Framework**: We developed a novel fully discrete algorithm that addresses a long-standing open problem: achieving the approximation guarantees of continuous methods for matroid constraints while maintaining the computational efficiency and simplicity of discrete algorithms. Our key insight comes from analyzing why continuous methods succeed: they implicitly use a two-loop structure with fractional increments. Block-Greedy translates this insight into a discrete setting by processing "blocks" of elements in each iteration, where the inner loop follows partition matroid analysis and the outer loop follows cardinality constraint analysis. This approach achieves optimal approximation ratios without continuous relaxation, resulting in both improved approximation guarantees compared to existing discrete algorithms and superior query complexity compared to continuous methods, particularly for Submodular Cover with fairness constraints.

- **Nonmonotone Conversion Algorithm**: We developed the first conversion algorithm for nonmonotone submodular cover by employing more delicate concentration inequality analysis that avoids truncated objectives (which only work for monotone functions). This extends the conversion framework to settings previously beyond its reach.

**3. Query Complexity Dependence on $c_{\max}/c_{\min}$ (Reviewer hyjR)**

We acknowledged that the $O(\log(c_{\max}/c_{\min}))$ factor in query complexity can be large but explained that this dependence is inherent to the conversion framework and appears in prior work. Critically, our dependence is logarithmic rather than polynomial in the ratio, keeping runtime polynomial when $c_{\max} \leq O(\exp(\text{poly}(n))) \cdot c_{\min}$, which holds in most practical scenarios.

**4. Experimental Validation (Reviewer 3RsE)**

Following the reviewer's suggestions, we added new experiments (Figure 3 in Appendix) demonstrating the impact of varying $\alpha$ on query complexity, runtime, and feasibility. These results confirm Block-Greedy's superior budget efficiency compared to baseline methods. We also clarified that similar function values across methods are expected behavior, as all algorithms are designed to achieve the $(1-\epsilon)\tau$ theoretical guarantee.

**5. Positioning Relative to Submodular Maximization Literature (Reviewer 2zuP)**

We expanded our related work discussion to clarify how SMKP differs from existing formulations. Specifically, SMKP enforces per-partition knapsack constraints (rather than a single global constraint), fundamentally distinguishing it from prior fairness-aware submodular maximization work that combines global knapsack constraints with per-partition cardinality bounds.

**Summary**: Three reviewers rated our paper 6 (marginally above acceptance threshold) with confidence levels of 3-4, while one reviewer rated it 4 (marginally below acceptance threshold) with confidence 3. All reviewers positively acknowledged our theoretical contributions and practical validation, with the primary concerns centered on presentation clarity and positioning relative to existing work, both of which we have substantially addressed in our revision.

We believe our comprehensive responses and manuscript revisions have strengthened the paper significantly, and we hope you will find it suitable for acceptance at ICLR 2026.

---

### Meta-Review · Area_Chair_8Rda · 2025-12-28

**Summary:**

The reviews are around the borderline, and somewhat towards rejection. The main concern is the strength of the result, and the lack of technical novelty, especially compared with existing works. Overall, I stand behind the reviewers and suggest to reject the paper.

**Reviewer Concerns:**

The rebuttal attempts to address the novelty issue. The authors mentioned that they resolve some well known open question, however, no clear reference is given and it is unclear to what extent it is a major open question in the field. The authors also mention many detailed points to justify the technical novelty, but they seem to be very specific; some of these steps are "the first", but they seem to follow from existing frameworks in a relatively straightforward way.

**Reviewer Scores:**

I think that the reviewers would keep the score unchanged.

---

### Decision · Program_Chairs · 2026-01-26

Reject